Citation: *Molecular Systems Biology* 9:695
www.molecularsystemsbiology.com

# Natural sequence variants of yeast environmental sensors confer cell-to-cell expression variability

Steffen Fehrmann[1], Hélène Bottin-Duplus[1], Andri Leonidou[1], Esther Mollereau[1], Audrey Barthelaix[1], Wu Wei[2], Lars M Steinmetz[2] and Gaël Yvert[1],*

[1] Laboratoire de Biologie Moléculaire de la Cellule, Ecole Normale Supérieure de Lyon, CNRS, Université Lyon 1, Lyon, France and [2] Genome Biology Unit, European Molecular Biology Laboratory, Heidelberg, Germany
* Corresponding author. Laboratoire de Biologie Moléculaire de la Cellule, Ecole Normale Supérieure de Lyon, CNRS, 46 Allée d'Italie, Lyon F-69007, France. Tel.: +33 4 72 72 87 17; Fax: +33 4 72 72 80 80; E-mail: Gael.Yvert@ens-lyon.fr

**Living systems may have evolved probabilistic bet hedging strategies that generate cell-to-cell phenotypic diversity in anticipation of environmental catastrophes, as opposed to adaptation via a deterministic response to environmental changes. Evolution of bet hedging assumes that genotypes segregating in natural populations modulate the level of intraclonal diversity, which so far has largely remained hypothetical. Using a fluorescent $P_{met17}$-GFP reporter, we mapped four genetic loci conferring to a wild yeast strain an elevated cell-to-cell variability in the expression of *MET17*, a gene regulated by the methionine pathway. A frameshift mutation in the Erc1p transmembrane transporter, probably resulting from a release of laboratory strains from negative selection, reduced $P_{met17}$-GFP expression variability. At a second locus, *cis*-regulatory polymorphisms increased mean expression of the Mup1p methionine permease, causing increased expression variability in *trans*. These results demonstrate that an expression quantitative trait locus (eQTL) can simultaneously have a deterministic effect in *cis* and a probabilistic effect in *trans*. Our observations indicate that the evolution of transmembrane transporter genes can tune intraclonal variation and may therefore be implicated in both reactive and anticipatory strategies of adaptation.**
*Molecular Systems Biology* **9**:695; published online 8 October 2013; doi:10.1038/msb.2013.53
*Subject Categories:* metabolic and regulatory networks; cellular metabolism
*Keywords:* bet hedging; complex trait; methionine; noise in gene expression; QTL

## Introduction

The genetics of adaptation to environmental conditions are usually apprehended from a deterministic point of view: rare or modified alleles get fixed in a population because they improve the fitness of individuals carrying them. Countless examples illustrate this general Darwinian mechanism of living systems. When environmental conditions fluctuate or change unpredictably, this strategy assumes a reactive mechanism: adapted individuals are those that perceive and respond rapidly to the new conditions. A number of observations support the existence of an alternative and complementary strategy that is probabilistic and based on anticipation. In this case, genotypes may be selected if they confer phenotypic diversity among isogenic individuals carrying them, because some of these carriers may be adapted 'in advance', 'by chance'. This strategy is often referred to as bet hedging, because it meets the original definition of a gain of geometric mean fitness over generations, at the cost of decreasing the arithmetic mean fitness (Lewontin and Cohen, 1969; Simons, 2011). Diversifying phenotypes may indeed hedge the bets: it has an immediate cost but can be rewarding in case of a catastrophe. The potential advantage of phenotypic diversification in fluctuating environments has been discussed and was shown to surpass the reactive strategy under certain timings of fluctuations (Kussell and Leibler, 2005; Acar *et al*, 2008; Stomp *et al*, 2008). The theory that evolution can act by bet hedging is attractive for two reasons. First, numerous cases of phenotypic diversification have been described that can confer persister individuals after environmental stress. This ranges from the persistence of bacteria after ampicillin treatment (Balaban *et al*, 2004), of rare yeast cells after severe heat stress (Levy *et al*, 2012), to the elevated variance in germination time of isogenic plant seeds (Simons and Johnston, 2006) and to the asymmetric distribution of carbon stocks among daughter cells of starved *S. meliloti* (Ratcliff and Denison, 2010). In addition, individual cells are not equally responsive to stress (Ni *et al*, 2012) or transcriptional induction (Robert *et al*, 2010), and this heterogeneity can greatly diversify the population response to environmental changes. Second, various types of mutations may change intraclonal phenotypic diversity. For a discrete adaptive trait, a mutation may change the rate of phenotypic switching between adapted and nonadapted states, for example, by changing the efficiency of a positive feedback loop in a gene regulatory network. For a quantitative adaptive trait,

mutations may act on various statistical properties of the trait values among carrier individuals, such as variance or skewness. This may happen by affecting feedback controls, by changing cooperativity at promoters (Becskei *et al*, 2005) or via the modulation of chromatin dynamics (Raser and O'Shea, 2004), transcriptional elongation (Ansel *et al*, 2008) or translation efficiency (Guido *et al*, 2007).

Most genetic studies of phenotypic switching and biological noise have relied on artificial manipulations made in the laboratory. One remarkable example was the experimental evolution of dimorphism in bacteria: extreme selection directly applied on phenotypic switching could fix mutations causing it (Beaumont *et al*, 2009). But what about natural alleles that segregate in natural populations and through which evolution takes place? Do they confer different levels of phenotypic diversity? If they do, then selection for bet hedging may happen in the wild. Otherwise, such a selective mechanism would first require a step where genotypes generating higher biological noise or modifying phenotypic switches appear in the population. A handful of examples illustrate that natural DNA polymorphisms can confer different levels of phenotypic heterogeneity. An important one is the fact that elevated developmental asymmetry (stochastic differences between left and right body parts) can be fixed using supervised crosses between natural fly stocks (Carter and Houle, 2011). This observation echoed previous reports on natural fly genotypes affecting the sensitivity to environmental variation in bristle number (Mackay and Lyman, 2005). Similar observations were made on snails (Ros *et al*, 2004) and maize (Ordas *et al*, 2008) where environmental trait variance was not uniformly seen across genotypes. A concordant example is the different degree of cell–cell variability in cellular morphology among wild yeast strains (Yvert *et al*, 2013). In humans, two genotypes have been linked to interindividual variation. The *Nf1* heterozygous mutation, which causes neurofibromatosis type 1 with varying levels of penetrance, was shown to increase morphological variability among cultured melanocytes from the same donor (Kemkemer *et al*, 2002). More recently, the *FTO* gene was not only associated with obesity, but also with interindividual variation of body mass index (Yang *et al*, 2012). More exhaustively, a genomic study of interindividual variation in plant metabolic and transcriptomic variation identified numerous genetic loci modulating this variability. Remarkably, the authors were able to validate the effect of natural *ELF3* alleles on the variation of the period of circadian transcriptional oscillations (Jimenez-Gomez *et al*, 2011). These examples illustrate that, indeed, natural alleles can confer different degrees of phenotypic noise and may therefore be the subject of selection under appropriate environmental fluctuations.

The existence of these genetic effects suggests that adopting a nondeterministic point of view may sometimes help understand the genetics of complex traits. To refine the usual concept of a quantitative trait locus (QTL) that affects the mean trait value of individuals, we propose to define a probabilistic trait locus (PTL) as a locus that changes the probability that an individual expresses a given trait value in a given genomic and environmental context (Box 1). Under this definition, all QTLs are PTLs but the reverse is not true: a PTL allele may increase

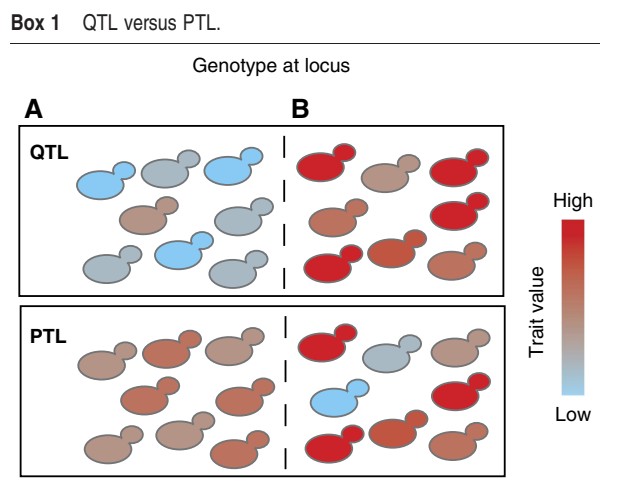

**Box 1** QTL versus PTL.

Genotype at locus

A quantitative trait locus (QTL) increases the trait value in individuals carrying a specific genotype at the locus. In contrast, a probabilistic trait locus (PTL) can change the probability that an individual displays a given trait value without necessarily changing the mean trait value of all individuals carrying the genotype. In the example shown, individuals with a B allele at the PTL are more likely to display extreme trait values than individuals with the A allele.

the chances to observe individuals with extreme phenotypes (e.g., with either low or high trait values) without necessarily changing the mean trait value of all carriers. A PTL may also change the transition rate between two phenotypic states without modifying the overall proportion of individuals being in one state at a given time. Note that many QTLs have deterministic effects that manifest only in specific genetic or environmental contexts, and therefore only in some individuals. In this case, the locus does not need to be called a PTL because its effect is not inherently probabilistic. Finally, the probabilistic effect of a PTL may be modulated by nonadditive interactions with the genetic background or the environment, just as the deterministic effect of a QTL. Identifying PTL in outbred populations presents two major difficulties. First, if individuals differ at many loci and all have their environmental history, one can hardly distinguish whether the variable expressivity of the phenotype is due to hidden genetic or environmental factors or to a probabilistic effect of the locus. In this regard, model organisms such as recombinant inbred lines of animals or plants are useful: they can be grown in replicates under controlled environmental conditions. The second limitation is the detection power of PTL. At similar power, detecting differences in variance or other high-order moments of a statistical distribution requires larger samples than detecting differences in mean.

One of the most convenient frameworks for recording very large samples and therefore estimating intraclonal phenotypic distribution accurately is the yeast *S. cerevisiae*. When the physiological state of cells is reported by fluorescence, flow cytometry offers high-throughput acquisitions on large populations of isogenic individuals. In particular, the response to environmental changes can be estimated. Yeast cells are very sensitive to variations in available sources of sulfur, which can be organic (methionine, cystein) or inorganic. Genetic screens based on methionine auxotrophy have decrypted the

regulatory network underlying the metabolism of sulfur assimilation (Thomas and Surdin-Kerjan, 1997). Many genes, such as *MET17* that codes for a homocysteine synthase, are repressed in the presence of methionine. A key component of this response is the Met4p transcriptional activator (complexed with Cbf1p and Met28p) that promotes gene activation but only at low intracellular concentrations of *S*-adenosyl methionine (AdoMet) when it is not degraded by the action of the SCF(Met30) ubiquitin ligase complex (Rouillon *et al*, 2000). Another important regulator is the Met31p/Met32p transcriptional regulator that prevents the transcription of *MET17* while promoting expression of other responsive genes (such as *MET3* and *MET14*) at high intracellular levels of AdoMet (Blaiseau *et al*, 1997).

Using a fluorescent reporter of *MET17* expression ($P_{met17}$-GFP), we previously showed that cell–cell variation in expression could be treated as a complex genetic trait (Ansel *et al*, 2008). This provides an ideal model system to identify expression PTLs (ePTLs) affecting the statistical properties of the activation of a regulatory pathway. Although this initial study demonstrated the segregation of cell–cell variability across natural genotypes, several limitations prevented a fully informative description (Ansel *et al*, 2008). First, the fact that cell–cell variability in expression was highly correlated with mean expression values led to identify loci that proved to be eQTLs, and not strictly ePTLs, except for one locus. Second, this strict ePTL locus was caused by a *ura3Δ0* mutation that was artificially created in the laboratory for auxotrophic selection purposes. It was therefore informative on the mechanisms generating cell–cell variation but not on the nature of ePTLs segregating in the wild. Finally, this locus explained only a fraction of the difference in cell–cell variation between two genetic backgrounds, leaving all additional putative ePTLs unidentified. We present here the genetic mapping of four of these additional ePTLs, the characterization of their specificity and the identification of natural sequence polymorphisms underlying two of them. This allows us to draw conclusions on the properties of natural alleles modulating intraclonal phenotypic diversification in the wild.

## Results

The term 'cell–cell variability' is used here to define cell-to-cell differences in the level of expression of a gene between isogenic cells grown in a common environment. When single-cell expression levels are recorded by flow cytometry, cell–cell variability can be quantified by the coefficient of variation (CV = s.d. divided by mean) of expression among the population of cells. Note that low expression levels are known to correlate with high CV values, which is consistent with the elevated stochasticity seen in systems harboring low numbers of regulatory molecules.

### Genetic introgression of high cell–cell variability

We previously showed that cell–cell variability in the expression of $P_{met17}$-GFP differed between wild yeast backgrounds (Ansel *et al*, 2008). It was particularly different between two unrelated *S. cerevisiae* strains frequently used as a model for complex trait dissection (Ehrenreich *et al*, 2009). Strain BY, isogenic to S288c, is used as a reference in many laboratories (Brachmann *et al*, 1998) and strain RM derives from a vineyard isolate (Brem *et al*, 2002). In our initial study, we showed that cell–cell variability in $P_{met17}$-GFP expression segregated as a quantitative trait in the BY × RM cross. The *ura3Δ0* mutation, which had been artificially introduced in RM shortly after its isolation, accounted for ~37% of the increased variability observed in RM (Ansel *et al*, 2008). Elevated variability of $P_{met17}$-GFP expression remained among RM cells after 'curing' the *URA3* gene (Figure 1A). Thus, the natural genetic loci accounting for the interstrain difference remained to be identified.

Here we describe the mapping of these loci by introgression. We chose this method because it allows uncoupling CV from mean expression. In the BY × RM cross, treating CV values directly as the trait of interest previously led us to identify two loci changing mean expression (eQTLs). These loci also increased cell–cell variability of expression but simply as a consequence of a lower mean expression (Ansel *et al*, 2008). We reasoned that if the elevated CV values of RM could be introgressed in BY without affecting mean expression, then the genetic sources of cell–cell variability *per se* could be identified. We therefore performed successive backcrosses, selecting Ura+ segregants of high cell–cell variability and unaffected mean at every generation. Three independent lineages of seven generations produced strains W7, X7 and Z7 that were Ura+ and retained only ~1% of the RM genome (Figure 1B). To analyze the properties of $P_{met17}$-GFP expression in these strains, we cultured them in parallel with a BY control, first in a repressive condition (1 mM methionine) and then over a range of decreasing methionine concentrations (200 to 0 µM) representing moderate to full activation of the MET17 promoter. All three strains displayed elevated cell–cell variability as compared with BY. Both X7 and Z7 displayed similar mean expression as BY, whereas strain W7 showed a reduced mean expression (Figure 1C). To determine if the increased cell–cell variability of W7 was simply a by-product of its lower mean expression, we fitted a linear model of CV versus mean dependence on the BY values and examined whether W7 samples deviated from this model. This was clearly the case, with CV values significantly higher than expected from the model (Figure 1C). The same was true for X7 and Z7 as well. Thus, at a given mean expression level, all three strains displayed higher cell–cell variability than BY.

### Identification of multiple ePTLs controlling cell–cell variability in $P_{met17}$-GFP expression

If elevated cell–cell variability of W7, X7 and Z7 is because of genotypes inherited from the RM genome, then all loci of RM genotype in these strains are potential ePTL candidates. We therefore sequenced the genome of these strains to determine which parts originated from RM. Paired-end sequencing of genomic DNA from RM, W7, X7 and Z7 was obtained on an Illumina Genome Analyzer IIx, producing ~45× coverage of each strain. Reads were aligned on the S288c genome. Using the RM data set, we derived 42 794 SNPs that we then used to detect RM genotypes in the introgressed strains. A total of

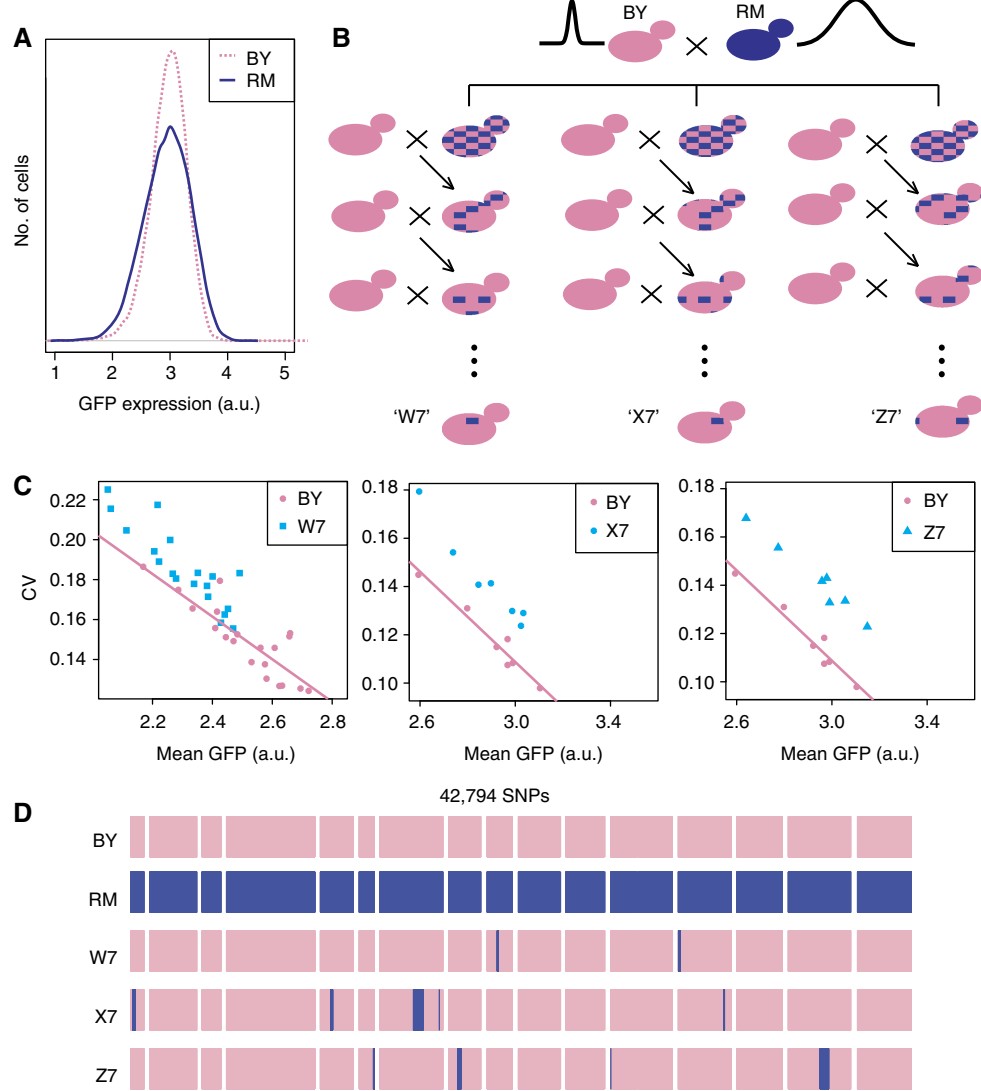

**Figure 1** Genetic introgression of elevated cell–cell variability in gene expression. (**A**) Representative distributions of GFP expression levels obtained by flow cytometry on populations of BY cells (strain GY51, pink) and RM cells (strain GY601, blue). Fluorescent values were corrected for cell size as indicated in the Materials and methods. Note that strain GY601 did not carry the original *ura3Δ0* mutation previously associated with elevated noise, which implies that the difference in variance is because of other sources of variability. (**B**) Introgression design. At every generation, a spore displaying elevated variability was backcrossed with BY. Blue patches denote remaining fractions of the genome originating from RM. (**C**) Characteristics of single-cell P$_{met17}$-GFP expression in the three strains resulting from introgression. Strains GY51 (BY), W7, X7 and Z7 were cultured, induced at decreasing concentrations of methionine (see Materials and methods) and analyzed by flow cytometry. Each dot represents at least 10 000 cells. Pink line: linear model fitted on the BY samples. All three introgressed strains display elevated cell–cell variability as compared with BY. (**D**) Whole-genome resequencing of the final introgression strains. Each horizontal band represents one chromosome from 1 (left) to 16 (right). SNP positions are printed as vertical bars, in pink when their genotype is BY and in blue when genotype is RM.

11 regions of RM origin were found in the three strains W7, X7 and Z7, with at least two regions introgressed in each one of them (Figure 1D and Table I). Two regions were single SNPs, likely representing sequencing errors or mutations in our BY strain as compared with the reference S288c genome. They were not considered further. All nine remaining regions were of RM genotype in only one of the three strains. At first sight, this was surprising. The introgression design being the same in all lineages, we expected to find common RM loci conferring high cell–cell variability in all of them. However, given our previous observation of polygenicity, this could simply be due to random drift and bottlenecks. At every generation, we chose one spore out of only ∼40. If, for example, six loci account for

the elevated variability of RM, then only one spore out of $2^6 = 64$ would harbor all genetic determinants. In this case, at any of the seven selection steps, measuring expression CV in only 40 spores may have led us to focus on a spore that retained only a subset of the loci of interest. And this subset may differ from one lineage to another.

We therefore considered all remaining nine regions as ePTL candidates and tested them directly on additional data sets. We crossed the seventh generation introgressed strains once more with BY, and performed linkage analysis on the resulting segregants. For example, crossing Z7 with BY generated a set of random spores that were genotyped at loci VI-263987, VIII-160869 and XV-625582, and these spores were analyzed

**Table I** Introgressed loci

| Strain | Chromosome | From[a] | To[a] | Marker position[a] | Validation *P*-value | ID |
|---|---|---|---|---|---|---|
| W7 | IX | 193 200 | 203 412 | 199 365 | 0.299 | — |
| W7 | XIII | 13 365 | 41 051 | 40 087 | 0.00928 | ePTL13 |
| X7 | I | 27 811 | 67 477 | 47 045 | 0.0388 | ePTL1 |
| X7 | V | 191 817 | 211 558 | 203 237 | 0.161 | — |
| X7 | VII | 577 844 | 745 593 | 599 377 | 6.66e − 05 | ePTL7 |
| X7 | VII | 1 026 099 | 1 026 099 | —[b] | —[b] | —[b] |
| X7 | XIII | 799 196 | 807 742 | 807 167 | 0.554 | — |
| Z7 | VI | 256 480 | 270 099 | 263 987 | 0.242 | — |
| Z7 | VIII | 160 589 | 212 237 | 160 869 | 2.41e − 11 | ePTL8 |
| Z7 | XII | 5741 | 5741 | —[b] | —[b] | —[b] |
| Z7 | XV | 563 573 | 703 057 | 625 582 | 0.4 | — |

[a]Position on the S288c genome (BY).
[b]The detected region was a single SNP and therefore not considered further.

by flow cytometry. To specifically test for an effect on CV and not mean expression, we first conditioned CV on mean expression level before using it as a trait for linkage (see Materials and methods). The rejection of one candidate locus and the validation of another one are shown as examples in Figure 2A and B. As indicated in Table I, four of the nine loci were validated as ePTLs and each introgression lineage contributed at least one validated ePTL.

### Three ePTLs recapitulate the high cell–cell variability of the wild RM strain

Strains W7, X7 and Z7 each contained several introgressed loci. To precisely estimate the individual effect of each ePTL, we built strains that were isogenic to BY except for a single locus of interest, and we measured $P_{met17}$-GFP cell–cell variability in these strains at various methionine concentrations (Figure 2C–F). We quantified the gain of cell–cell variability in the presence of each locus by computing the difference between observed CV and its expected value from a linear model fitted on BY samples (Figure 2H, see Materials and methods). The statistical significance of this gain was then tested by a one-tail Student's *t*-test of departure from zero. The *ePTL1* locus showed a CV increase that was statistically significant ($P = 0.00012$) but very mild (0.003 only). In contrast, expression CV increased markedly in response to *ePTL7*, *ePTL8* or *ePTL13* with gains of 0.014, 0.018 and 0.019, respectively, in the corresponding strains (Figure 2H). In comparison, the BY/RM difference in CV after removing the effect of the artificial *ura3* mutation was 0.031 (Figure 2H). Hence, each of these three loci increased cell–cell variability to an extent of approximately half of the parental difference.

To see how much cell–cell variability could be gained when cumulating all three loci with strong effect, we constructed a strain isogenic to BY except at these loci, which were of RM genotypes. We also brought a fourth locus on chromosome XV (Table I) from RM in this strain because we suspected it to be an additional ePTL, which later proved not to be the case. This strain displayed higher cell–cell variability than any of the strains harboring a single ePTL locus from RM. It also showed a decreased mean expression as compared with both BY and

RM (Figure 2G), which was consistent with the small decrease in mean expression seen in each of the three strains harboring a single ePTL from RM. The strain cumulating all three ePTLs showed a CV gain of 0.049, which is comparable to the sum of individual gains conferred by each locus (0.051). Interestingly, when introduced together into BY, the three ePTLs conferred higher cell–cell variability than the level seen in RM (Figure 2G). We previously suspected that BY contributes alleles conferring elevated variability because of the transgressive segregation of CV values in the BY–RM cross (Ansel *et al*, 2008). Now that the effect of ePTL in the BY context is shown to surpass the variability seen in RM, we can propose that these BY alleles are compensated by one or more of the three ePTLs identified. The fact that the elevated cell–cell variability of RM was recapitulated, and even surpassed, by combining three ePTLs suggests that we successfully mapped the major sources of it from RM.

### Specificity versus pleiotropy of ePTL effects

Genetic variants affecting cell–cell variability in the expression of a reporter gene may do so specifically or not. Specificity is expected if an ePTL changes the transcriptional control at the $P_{met17}$ promoter but not at other promoters, or if it modulates the maturation or degradation of GFP but not of other proteins. Alternatively, an ePTL may affect cell–cell variability in expression of many unrelated genes. This is probably the case for the *ura3* mutation as well as *dst1* or other mutations impairing transcriptional elongation (Ansel *et al*, 2008), because all protein-coding genes require proper dynamics of uracil supply and efficient release from elongation pausing. To determine whether the three major ePTLs identified here were specific or pleiotropic, we constructed a reporter that was unrelated to the methionine metabolism. We chose the promoter of the *ACT1* gene, which codes for actin, and placed it upstream the GFP coding sequence. We integrated the construct in a BY strain at the *HIS3* locus, as we did above for $P_{met17}$-GFP. We then used the introgression strains that are isogenic to BY except for one ePTL. We crossed each of these strains with the $P_{act1}$-GFP strain, which generated segregating populations in which linkage could be analyzed. To test the above expectation of pleiotropic cell–cell variability in the context of *ura3* or *dst1* mutation, we also crossed the $P_{act1}$-GFP strain with a *ura3* and a *dst1* mutant, respectively. In each cross, we tested for a possible linkage between the locus of interest (*ePTL*, *ura3* or *dst1*) and the CV of $P_{act1}$-GFP expression. Spores were generated and genotyped at the locus of interest. They were then cultured in the same condition as above and single-cell GFP expression was quantified by flow cytometry. No association was found between any of the three ePTLs and the CV of $P_{act1}$-GFP expression (Figure 3A–C). In contrast, both *ura3* and *dst1* mutations were associated with increased CV of $P_{act1}$-GFP, confirming their expected pleiotropy (Figure 3D and E). None of the loci affected $P_{act1}$-GFP mean expression, underlining that differential cell–cell variability is not a by-product of differential mean expression. We conclude that artificial mutations causing impairments in uracil supply or transcriptional elongation generate elevated cell–cell variability

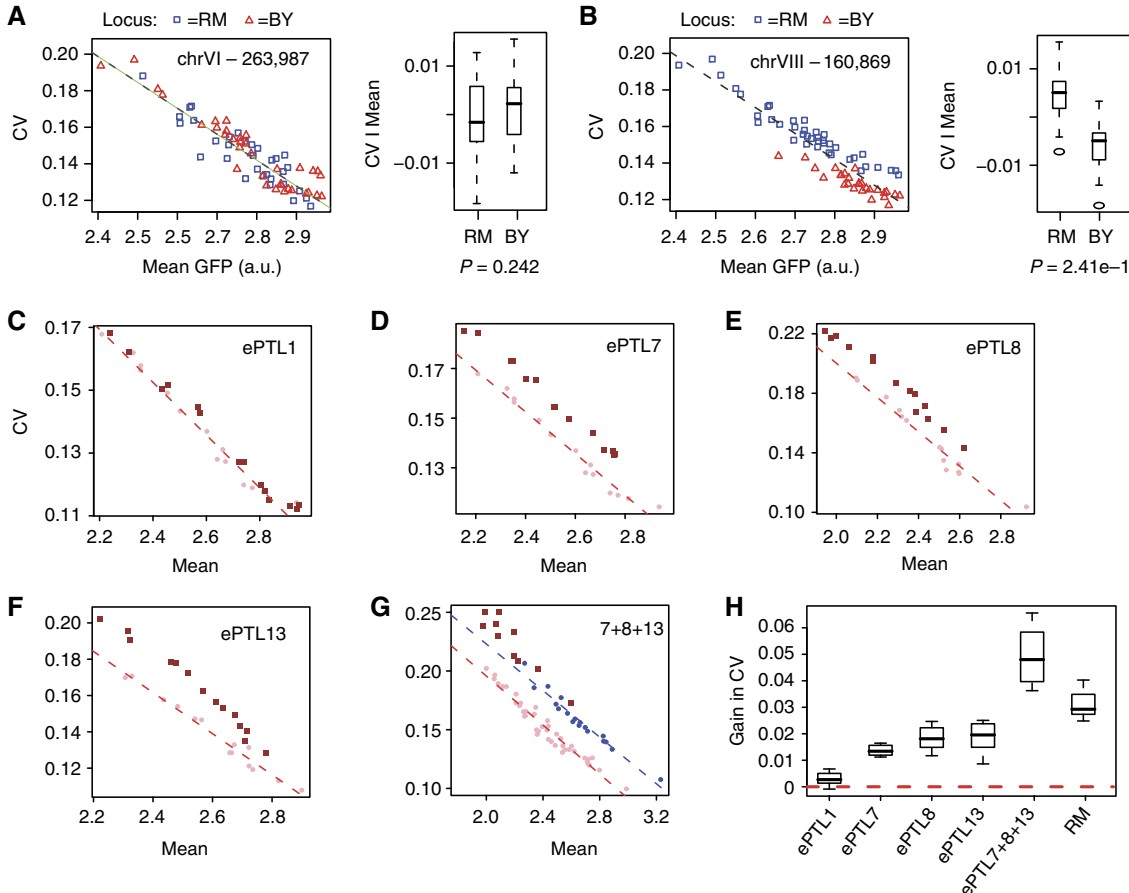

**Figure 2** Validation of some ePTL candidates. (**A**, **B**) Examples of rejection (**A**) and validation (**B**) of a candidate locus. Strain Z7 was crossed with BY, random spores were isolated and analyzed by flow cytometry after 2 h of induction at 50 μM methionine. Each dot represents 10 000 cells amplified from one spore. Fluorescent values were corrected for cell size as indicated in the Materials and methods. Spores were genotyped at position 263 987 on chromosome VI (**A**) and position 160 869 on chromosome VIII (**B**). Genotypes are indicated by symbols: red triangle, BY and blue square, RM. Dashed line: linear regression fitted on all data points. Box plots on the right represent the residuals (deviation from linear model) as a function of genotype. *P*, significance of a Wilcoxon Mann--Whitney test on the null hypothesis of no difference between the two sets of residuals. (**C**–**F**) Effects of four ePTLs. Cell–cell variability versus mean dot plots showing the extent of CV increase in strains isogenic to BY except for one ePTL locus originating from RM. These strains were GY926 (**A**), GY927 (**B**), GY919 (**C**) and GY915 (**D**) and are represented as dark squares. ePTLs are numbered by their chromosomal context, as in the last column of Table I. Pink dots indicate BY. Samples were cultured in a range of methionine concentrations. Each symbol represents 10 000 cells, fluorescent values were corrected for cell size as described in the Materials and methods. Red dashed line, linear regression model fitted on BY samples. (**G**) Same representation with strain GY943 that carried three ePTLs from RM. Blue dots indicate RM strain GY601. Blue dashed line, linear regression fitted on RM samples. (**H**) Box plot summarizing gains of CV as compared with BY. Using the same data as in **C**–**G**, the gain of CV in each sample was calculated as the difference between the observed CV value and the expected CV value given the linear model fitted on BY.

with no specificity, whereas natural ePTLs can act specifically on the regulation of the methionine biosynthesis pathway.

## Fine mapping of an introgressed ePTL

We then sought to identify DNA polymorphisms responsible for differences in cell–cell variability. The introgressed region containing *ePTL8* was over 50 kb in size and contained 25 annotated protein-coding genes. Given this relatively large number, we decided to refine the mapping of *ePTL8*. We did this by generating a set of haploid strains that were all isogenic to BY except for portions of the region. As shown in Figure 4A, the region is flanked by two genes that can be used as selective markers. *THR1* codes for a homoserine kinase essential for threonine biosynthesis (Mannhaupt *et al*, 1990). The *COX6* gene codes for a subunit of cytochrome C oxidase essential for

respiration (Gregor and Tsugita, 1982). Thus, both *THR1* and *COX6* functional genes are required for growth on a synthetic medium lacking threonine and with glycerol as the sole carbon source (Gly, Thr − ). Using the strain in which *ePTL8* was introgressed, we constructed a diploid strain where the genome was BY/BY homozygous except at the *ePTL8* region, which was BY/RM heterozygous and where the RM and BY haplotypes contained a *cox6Δ*- and a *thr1Δ*-null mutation, respectively (Figure 4B). This diploid was sporulated, and millions of spores were plated on (Gly, Thr − ) selective plates. As expected, only a few colonies were able to grow. We verified that these were not diploid cells but haploids generated from meiotic spores and we mapped the expected recombination events by genotyping eight markers spanning the *ePTL8* region (Figure 4C). To see which part of the region showed highest genetic linkage with cell–cell variability, we cultured these strains as above and quantified P$_{met17}$-GFP expression

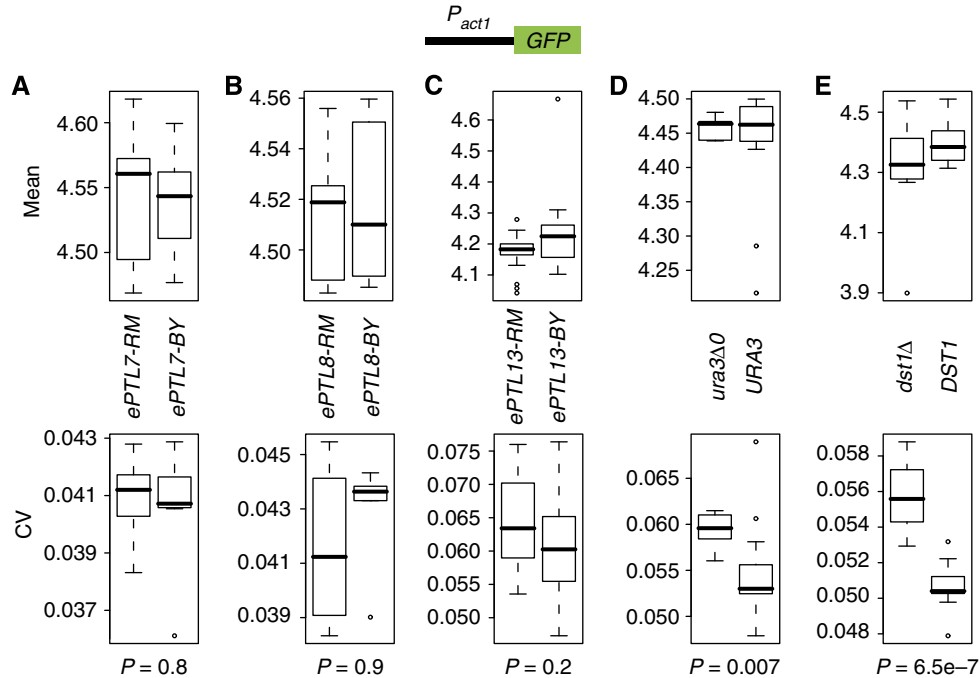

**Figure 3** Linkage analysis between ePTL and ACT1-GFP cell–cell variability. Strains carrying a single ePTL of $P_{met17}$-GFP variability were crossed with a BY strain carrying a $P_{act1}$-GFP reporter. Random spores carrying the $P_{act1}$-GFP but not the $P_{met17}$-GFP construct were isolated, amplified and cultured as above, analyzed by flow cytometry and genotyped at the segregating ePTL. Fluorescent values were corrected for cell size as described in the Materials and methods. Box plots indicate mean (upper panels) and CV (lower panels) of $P_{act1}$-GFP expression in spores carrying the BY or RM genotype at the ePTL. *P*, significance of a Wilcoxon Mann–Whitney test on the null hypothesis of no difference between the two sets of CV values. Crosses were GY935 × GY975 (**A**), GY930 × GY975 (**B**), GY91 × GY1156 (**C**), BY4719 × GY975 (**D**) and GY512 × GY975 (**E**).

by flow cytometry. Strains differed in both mean and CV of expression. Remarkably, the genotype at position 175 005 perfectly distinguished two low-variability strains from the others (Figure 4C). Interestingly, genotype at position 186 650 was associated with a difference in mean expression. Thus, generating strains that recombined within the 50 kb region allowed us to identify two functional subregions of it. The first one spanned 6 kb, between positions 171 530 and 177 601, and was perfectly associated with $P_{met17}$-GFP cell–cell variability. The second one was larger, from position 177 601 to 209 170, and was associated with $P_{met17}$-GFP mean expression.

## A frameshift mutation in the *Erc1p* transmembrane transporter decreases cell–cell variability of $P_{met17}$-GFP expression

The 6 kb region of *ePTL8* associated with $P_{met17}$-GFP cell–cell variability contained a single gene, *ERC1*. Previous studies supported a possible implication of this gene in the regulation of the *MET17* promoter. First, *erc1* mutants display increased resistance to ethionine, an antagonist of methionine (Shivapurkar *et al*, 1984; Shiomi *et al*, 1991). Second, overexpression of *ERC1* increases the intracellular concentration of AdoMet (Shiomi *et al*, 1995), a methyl donor directly derived from methionine and central in the metabolic pathway of sulfur amino acids (Thomas and Surdin-Kerjan, 1997). Finally, the BY and RM alleles of *ERC1* were shown to display different

transcriptional levels in BY/RM diploid hybrid cells. A slightly preferential expression of the RM allele indicated the presence of functional *cis*-regulatory polymorphisms in this gene (Ronald *et al*, 2005). The gene product of *ERC1* belongs to the MATE family of transmembrane transporters, which typically possess 12 transmembrane helices (Omote *et al*, 2006). We aligned the protein sequences of BY and RM and found a frameshift mutation in BY that truncated the last two predicted helices. When aligning protein sequences from other *S. cerevisiae* and *S. bayanus* strains, we saw that this mutation was specific to laboratory strains (Figure 4D) and to strain CLIB324, a bakery strain known to be genetically close to laboratory strains (Schacherer *et al*, 2009). To look for signs of selection, we computed the nonsynonymous over synonymous Ka/Ks ratio of the *ERC1* coding sequence, by comparing the RM variant with sequences from *S. paradoxus* and *S. bayanus*. This ratio was significantly low (0.05 and 0.025, respectively, $P < 10^{-60}$), suggesting that the protein sequence has been under purifying selection in the wild. Thus, the frameshift mutation of BY reflects a genetic defect acquired in laboratory strains, possibly by genetic drift after the ancestor was brought to the laboratory.

To directly test whether *ERC1* sequence polymorphism caused a difference in cell–cell variability of $P_{met17}$-GFP expression, we introduced the BY frameshift mutation in the genome of RM and we removed it from the genome of BY. Such allele-replacement manipulations in *S. cerevisiae* are traditionally done using integrative plasmids allowing selection (pop-in) and further counterselection (pop-out) (Scherer and Davis, 1979). This

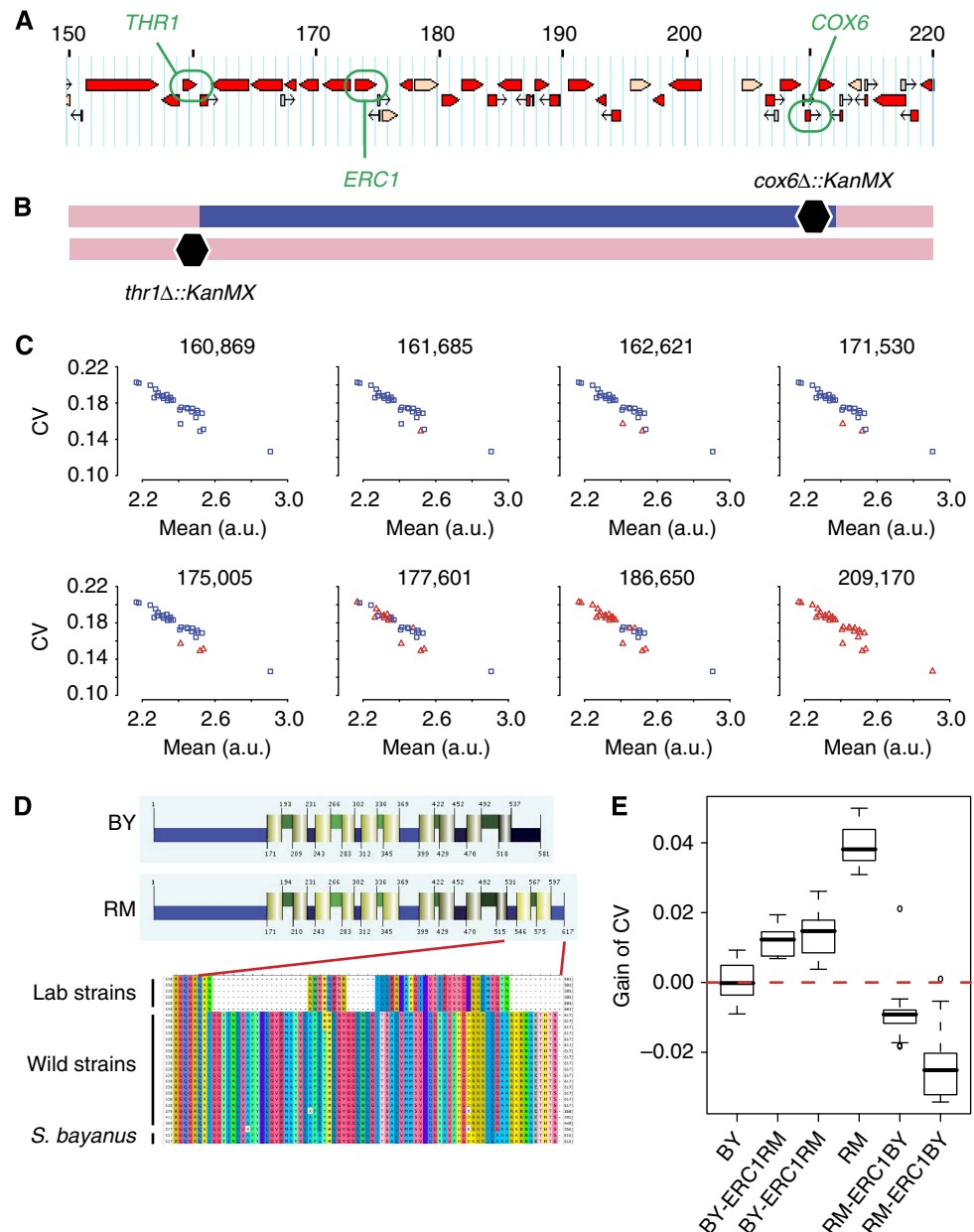

**Figure 4** Fine mapping of an ePTL polymorphism in the *ERC1* gene. (**A**) Physical map of the *ePTL8* introgressed locus. Positions are indicated in kb. (**B**) Genotype of GY939, the diploid strain used to generate spores that recombined within the introgressed region. Blue and pink regions represent RM and BY genotypes, respectively. The recessive *thr1Δ::KanMX4* mutation prevents spores from growing without threonine supplementation, and the recessive *cox6Δ::KanMX4* mutation prevents spores from growing on glycerol. Recombinants were obtained from a screen on glycerol plates lacking threonine. (**C**) Flow cytometry analysis of MET17-GFP expression in the recombinant spores obtained. Each dot represents 10 000 cells amplified from one spore, printed as a blue square or as a red triangle depending on whether the genotype at the indicated position was RM or BY, respectively. Fluorescent values were corrected for cell size as described in the Materials and methods. Marker position 175 005 discriminated low CV from high CV samples. (**D**) Predicted transmembrane domains in the Erc1p protein, as determined by Philius (Reynolds *et al*, 2008) from BY and RM sequences. The colored protein sequence alignment corresponds, from top to bottom, to *S. cerevisiae* strains BY4741, BY4742, S288c, W303, CLIB324 and FL100 known to be related to laboratory strains (Schacherer *et al*, 2009), *S. cerevisiae* strains CBS7960, JAY291, T73, EC1118, EC9-8, Sigma1278b, Kyokai7, UC5, T7, YJM269, LalvinQA23, AWRI1631, AWRI796, CLIB215, RM11-1a, YJM789, FostersB, VL3, Vin13 and FostersO that sample the wild diversity of the species, and two strains from the closely related species *S. bayanus*. The C-terminal frameshift mutation of BY is found exclusively in the laboratory strain group. (**E**) Allelic replacements of *ERC1* affect cell–cell variability of MET17-GFP expression. Strains GY1023 and GY1024 were independently derived from BY with *ERC1* replaced by the RM allele (*BY-ERC1RM*), and strains GY1019 and GY1020 were independently derived from RM with *ERC1* replaced by the BY allele (*RM-ERC1BY*). These strains were cultured in parallel of GY246 (BY) and GY53 (RM), induced at decreasing concentrations of methionine and their level of GFP expression was analyzed by flow cytometry. A linear model of CV versus mean was fitted to the BY samples and gain of CV was calculated as in Figure 2H. The CV of $P_{met17}$-GFP significantly increased when introducing the RM allele into BY ($P < 1e - 5$, Wilcoxon Mann–Whitney), and significantly decreased when introducing the BY allele into RM ($P < 1e - 7$, Wilcoxon Mann–Whitney).

requires the cloning of genomic fragments into such plasmids, which is sometimes not convenient when performed *in vitro*. Some authors developed alternative, cloning-free methods based on oligonucleotides (Storici and Resnick, 2006), but, at least in our hands, these methods were limited by their efficiency of integration in the genome. To bypass the limiting *in vitro* cloning steps of the plasmid-based strategy, we constructed a vector carrying centromeric and autonomous replication sequences (CEN/ARS) that are flanked by *LoxP* sites. This vector can be used in its replicative form for *in vivo* cloning by homologous recombination. The CEN/ARS sequence of the resulting construct can then be excised by the Cre recombinase to obtain an integrative version suitable for pop-in and pop-out (see Materials and methods). Using this strategy, we obtained two independent BY strains carrying an RM-like allele of *ERC1*, and two independent RM strains carrying a BY-like allele of *ERC1* (Supplementary Table S1). Importantly, one of these strains was modified only at the frameshift mutation and two SNPs immediately downstream, allowing to precisely evaluate the implication of this mutation. We analyzed $P_{met17}$-GFP expression in these strains by flow cytometry over a range of methionine concentrations and we computed how much cell–cell variability was gained or lost in response to the allelic replacement. All modified strains displayed a significant change of CV in the expected direction: the RM allele of *ERC1* increased cell–cell variability of $P_{met17}$-GFP expression when introduced into BY, to a level similar to the effect attributed above to *ePTL8*, and the BY allele decreased it when introduced into RM (Figure 4E). Remarkably, the effect was much more pronounced when manipulating the RM strain: introducing a BY allele of *ERC1* conferred an even lower CV than the value seen in BY, whereas introducing the RM allele into BY caused a moderate CV increase. This strongly argues for a genetic interaction between *ERC1* and additional genes. We saw above that the cumulative effect of *ePTL7*, *ePTL8* and *ePTL13* was higher than expected in the BY context, indicating the presence of BY alleles increasing variability (Figure 2G and H). Here we see that *ERC1* seems to interact with other loci of the genome, where a BY context does not amplify but attenuates the effect of the mutation. It is therefore likely that several epistatic interactions, sometimes antagonistic, take place between ePTL. Altogether, modifying specifically the *ERC1* genotype proved its effect on cell–cell variability of $P_{met17}$-GFP. Given the tight connection between *ERC1* and the methionine metabolic pathway, this result explains the observed specific and not pleiotropic effect of *ePTL8* (Figure 3B).

## *Cis*-regulatory polymorphisms in the *MUP1* methionine permease gene affect cell–cell variability of $P_{met17}$-GFP expression in *trans*

The two other major ePTLs were not flanked by genes that can facilitate the selection of recombinant spores. One of them, *ePTL7*, was a 168-kb region on chromosome VII containing ~80 genes. Knowing that the effect of *ePTL7* was not pleiotropic but somewhat specific to the regulation of the *MET17* promoter (see above), we searched among these genes

for annotations connected to the sulfur and methionine pathway. The *MUP1* gene stood out as an interesting candidate because it encodes a methionine permease (Isnard *et al*, 1996). If the efficiency or dynamics of methionine uptake differ between BY and RM, this may likely generate differences in the level or dynamics of *MET17* repression. In addition, *MUP1* was previously identified as a *cis*-eQTL in the BY × RM cross, suggesting a possible differential activity (Smith and Kruglyak, 2008). We therefore examined the possible implication of *MUP1* in the modulation of cell–cell variability of $P_{met17}$-GFP expression.

The BY and RM Mup1p proteins are 100% identical in sequence. However, several sequence polymorphisms resided in the promoter region of the gene. Three of these created transcription factor binding sites in RM that were not present on the BY allele (Figure 5A). Sequence alignment of the 750 bp promoter region from 29 strains of *S. cerevisiae* and other species indicated various combinations of the presence/absence of these sites among natural populations (Supplementary Figure S1). Hence, unlike for the *ERC1*

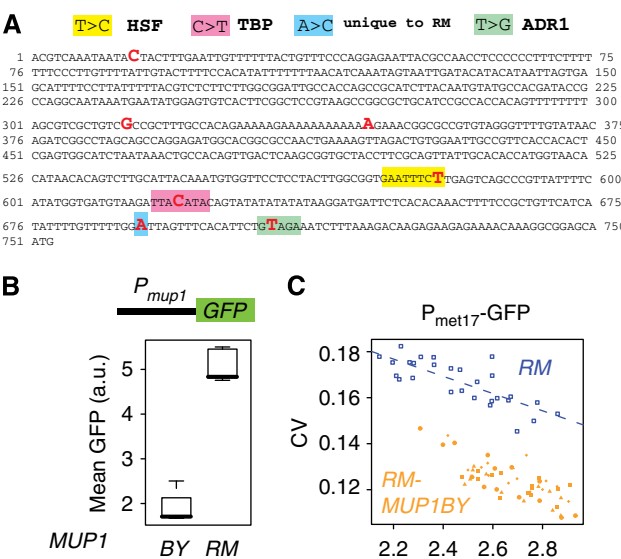

**Figure 5** *Cis*-regulatory variation in *MUP1* increases MET17-GFP cell–cell variability in *trans*. (**A**) BY/RM polymorphisms in the 750 bp promoter region of MUP1. The BY sequence is printed. Red, nucleotides differing in RM. Boxes colored yellow, pink and green correspond to binding sites present in RM but not in BY for transcription factors HSF, TBP and ADR1, respectively, with the nucleotide changes indicated above the sequence. The blue box corresponds to a SNP specifically found in RM and none of 24 other *S. cerevisiae* strains. (**B**) The RM allele of *MUP1* confers high promoter activity. The *MUP1* promoter from BY was cloned upstream the GFP reporter, and the construct was integrated in a BY strain to produce independent transformants strains GY1005, GY1006 and GY1007. The *MUP1* promoter from RM was processed similarly to produce strains GY1002, GY1003 and GY1004. All six strains were analyzed by flow cytometry in the same experimental conditions as above (2 h at 50 μM methionine). (**C**) Allele-replacement experiment. The genome of RM-derived strain GY53 was manipulated in order to replace the promoter region of *MUP1* by its corresponding BY sequence. Four independent strains were obtained (GY1205–1208). Their MET17-GFP expression was analyzed by flow cytometry at decreasing methionine concentrations. For comparison, the unmodified RM strain GY53 was cultured and analyzed in parallel (blue squares). Dashed line: linear regression fitted to RM samples. Each orange symbol represents one *MUP1* replacement strain.

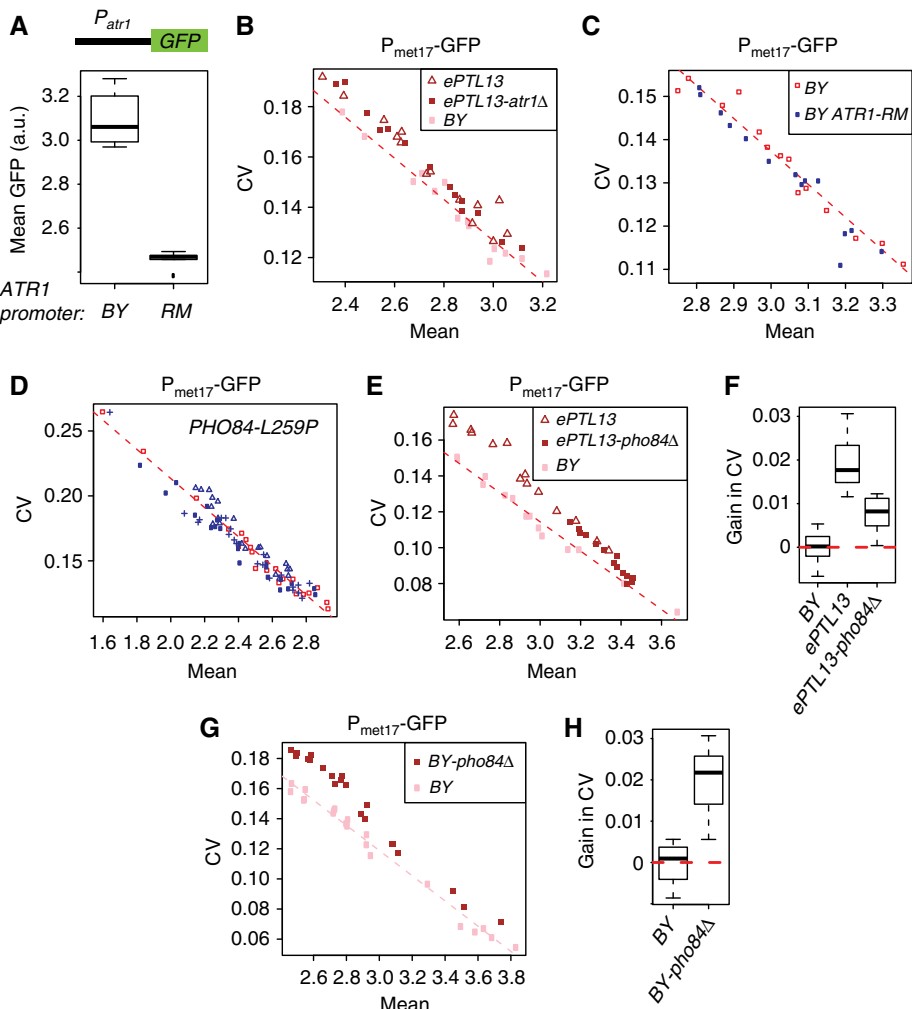

**Figure 6** Functional evaluation of *ATR1* and *PHO84* genes located at *ePTL13*. (**A**) The BY allele of *ATR1* confers high promoter activity. The ATR1 promoter from BY was cloned upstream the GFP reporter, and the construct was integrated in a BY strain to produce independent transformants strains GY1325 and GY1326. The ATR1 promoter from RM was processed similarly to produce strains GY1327 and GY1328. All four strains were analyzed by flow cytometry in triplicates in the same experimental conditions as above (2 h at 50 μM methionine). (**B**) Deletion of *ATR1* does not modify the contribution of ePTL13. Strains BY (pink squares, GY172), ePTL13 (brown triangles, GY915) and ePTL13-*atr1Δ* (filled brown squares, GY1309) were cultured in parallel, induced at decreasing concentrations of methionine and their level of GFP expression was analyzed by flow cytometry. Dashed red line: linear model of CV versus mean fitted to the BY samples. (**C**) No incidence of ATR1 allele replacement on $P_{met17}$-GFP expression variability. Strain GY1330 (blue), isogenic to BY except for the ATR1 promoter that was replaced by the RM allele, was cultured in parallel of strain GY246 (red) and analyzed as in (**B**). (**D**) No incidence of PHO84-L259P polymorphism on $P_{met17}$-GFP expression variability. Strains GY1306 (blue filled squares), GY1307 (blue triangles) and GY1308 (blue crosses) isogenic to BY except for the PHO84-L259P polymorphism were compared with BY strain GY172 (red) as in (**B**) and (**C**). (**E**) Deletion of PHO84 partly reduces the effect of ePTL13 and increases mean expression. Strain GY1310 (filled brown squares) carrying the ePTL13 locus from RM in which the *PHO84* gene was deleted was compared with BY (GY51, pink squares) and to strain GY915 (brown triangles), which carried an intact ePTL13 locus from RM. Dashed red line: linear model of CV versus mean fitted to the BY samples. (**F**) Quantification of variability changes in response to *pho84* deletion using the data presented in (**E**). Gains in CV were calculated as the deviations from the linear model fitted to BY samples. CV gains in ePTL13-*pho84Δ* were significantly reduced as compared with ePTL13 (Wilcoxon Mann–Whitney $P = 3.5 \times 10^{-7}$) but remained significantly higher than BY (Wilcoxon Mann–Whitney $P = 7 \times 10^{-6}$). (**G**) Deletion of *PHO84* in the BY context increases $P_{met17}$-GFP expression variability. Strain BY-*pho84Δ* (GY1296, brown squares) was compared with strain BY (GY51, pink squares) as in (**B**), (**C**) and (**E**). Dashed red line: linear model of CV versus mean fitted to the BY samples. (**H**) CV gains calculated on the data presented in (**G**). The increase in CV was highly significant (Wilcoxon Mann–Whitney $P = 3 \times 10^{-11}$).

frameshift mutation described above, laboratory strains did not present a particularly 'uncommon' *MUP1* haplotype. The only noticeable peculiarity was one SNP found only in RM. To test whether these BY/RM polymorphisms conferred different *MUP1* promoter activities, we cloned both promoter variants upstream the GFP coding sequence, and we integrated the resulting construct in a BY strain. Flow cytometry quantification indicated a marked difference, with ~2.5 times higher expression when using the RM promoter variant (Figure 5B).

This observation explains the *cis*-eQTL previously reported (Smith and Kruglyak, 2008). To determine whether these *cis*-regulatory polymorphisms in *MUP1* underlay *ePTL7*, we manipulated the genome of the RM strain and replaced its *MUP1* promoter by the BY variant. We obtained four independent strains all carrying the BY version of the promoter. We measured single-cell $P_{met17}$-GFP expression in these strains by flow cytometry at various methionine concentrations and compared them with the unmodified RM

strain. All four strains behaved consistently, showing a marked decrease in variability level as compared with RM (Figure 5C). By computing the deviation of the modified strains from a linear model fitted to RM samples, we estimated that the replacement of the *MUP1* promoter allele caused a CV reduction of 0.035. This effect is bigger than the gain of variability contributed by *ePTL7* in the BY genomic context (0.014, Figure 2H). As for *ERC1*, this supports the possibility that RM alleles at additional loci may modify the ePTL effect of *MUP1*. In conclusion, promoter activity tests and the direct manipulation of the *MUP1* promoter in its genomic context demonstrated that it is a *cis*-acting eQTL (RM allele increasing MUP1 transcript level) and a *trans*-acting ePTL (RM allele increasing $P_{met17}$-GFP expression variability).

## Functional analysis of the *ATR1* and *PHO84* genes located at *ePTL13*

The *ePTL13* locus did not contain any obvious candidate but we noticed two genes of interest that we tested for a possible effect on $P_{met17}$-GFP expression variability. *ATR1* encodes a transmembrane protein implicated in multidrug resistance (Kanazawa *et al*, 1988). Its mRNA expression is co-activated on boron stress with the expression of genes involved in amino-acid biosynthesis, in a GCN4-dependent manner (Uluisik *et al*, 2011), and *ATR1* was previously identified as a *cis*-eQTL in the BY × RM cross (Smith and Kruglyak, 2008). The amino-acid sequence of the Atr1p protein is identical between BY and RM. We reasoned that a regulatory polymorphism in the ATR1 promoter could affect cell–cell variability in *trans*, as in the case of *MUP1*. We therefore compared the activity of the BY and RM promoter alleles of *ATR1* by cloning these promoters in GFP-coding integrative constructs. The BY allele displayed a significantly stronger activity than the RM allele, confirming the earlier *cis*-eQTL result (Figure 6A). We then performed two functional experiments to test if *ATR1* polymorphisms affect $P_{met17}$-GFP expression variability. First, we deleted *ATR1* in a BY strain containing the introgressed *ePTL13* locus from RM. This alteration did not reduce the elevated cell–cell variability of this strain (Figure 6B). Thus, *ePTL13* does not need the presence of *ATR1* to increase $P_{met17}$-GFP expression variability. Second, we replaced the endogenous *ATR1* promoter of a BY strain by the RM allele. Again, this modification did not modify $P_{met17}$-GFP cell–cell variability (Figure 6C). These two experiments excluded *ATR1* as a gene responsible for the effect of *ePTL13*.

The second gene of interest at *ePTL13* was *PHO84*. This gene encodes a transmembrane transporter of inorganic phosphate (Bun-Ya *et al*, 1991) that also acts as a sensor to activate the protein kinase A (PKA) pathway (Giots *et al*, 2003). It was previously identified as a major QTL for the resistance to two polychlorinated aromatic drugs in the BY × RM cross, with a nonsynonymous SNP (L259P) causing this effect (Perlstein *et al*, 2007). We reasoned that sequence polymorphisms in this gene could modify the dynamics of phosphate transport or the triggering of PKA signaling, with possible consequences on $P_{met17}$-GFP expression variability. To directly test the possible implication of the L259P polymorphism, we

quantified $P_{met17}$-GFP variability in a BY strain carrying this SNP. No change was observed in comparison with BY (Figure 6D). To test the possible implication of other polymorphisms (coding or not), we deleted *PHO84* in the BY strain containing the introgressed *ePTL13* locus from RM. As seen in Figure 6E and F, this mutation had two consequences on $P_{met17}$-GFP regulation: methionine repression was less efficient, and cell–cell variability was reduced (even after accounting for the effect on mean expression). For comparison, we then monitored $P_{met17}$-GFP expression in a fully BY strain carrying the *pho84Δ* deletion. In this BY context, the consequence of the mutation was different: methionine repression was not impaired and, instead of decreasing, cell–cell variability markedly increased (Figure 6G and H). The fact that BY-*pho84Δ* and *ePTL13-pho84Δ* strains differ suggests that polymorphisms in other genes located at *ePTL13* modify the effect of *pho84Δ* deletion on the extent of methionine repression. The fact that *pho84Δ* increased variability in the BY strain but partially decreased it in the *ePTL13* strain suggests that BY and RM PHO84 alleles may have differential impacts on $P_{met17}$-GFP variability. In conclusion, these results support the implication of PHO84 on the modulation of $P_{met17}$-GFP variability and further investigations are needed to completely understand the contribution of this gene and other variants of the locus.

## Discussion

We mapped four natural loci contributing to cell–cell variability in the expression of a $P_{met17}$-GFP fluorescent reporter. Because these loci changed the statistical properties of single-cell gene expression level rather than its mean, we called them ePTLs. Three of these loci were enough to recapitulate the high cell–cell variability seen in the RM parental strain, and they did not affect variability in the expression of an unrelated gene. Allele-replacement experiments demonstrated that natural alleles at two genes, *ERC1* and *MUP1*, both coding for transmembrane transporters, increased cell–cell variability of expression in *trans*. Deletion of the *PHO84* phosphate transporter also affected variability in the expression of the reporter.

All four ePTLs were obtained by introgression mapping. This approach offered several major advantages over traditional schemes. First, it allowed us to decouple two highly correlated traits (mean expression and CV) so that genetic determinants of cell–cell variability *per se* could be identified. Second, its cost was modest because only three genomes were genotyped instead of a panel of segregants. Finally, in addition to the statistical detection of causative loci, the method produced organisms that were nearly isogenic to one strain except for these loci. These organisms enabled further functional analysis of the locus before the causative sequence polymorphism was identified. Here, the introgressed strains allowed us to test the specificity of ePTL effects, to reduce a candidate region down to a single gene and to perform informative gene-deletion experiments on specific candidate genes. Such introgressed organisms may be invaluable in various studies. They can help identify candidate genes and they can also help scientists decide whether or not the locus is

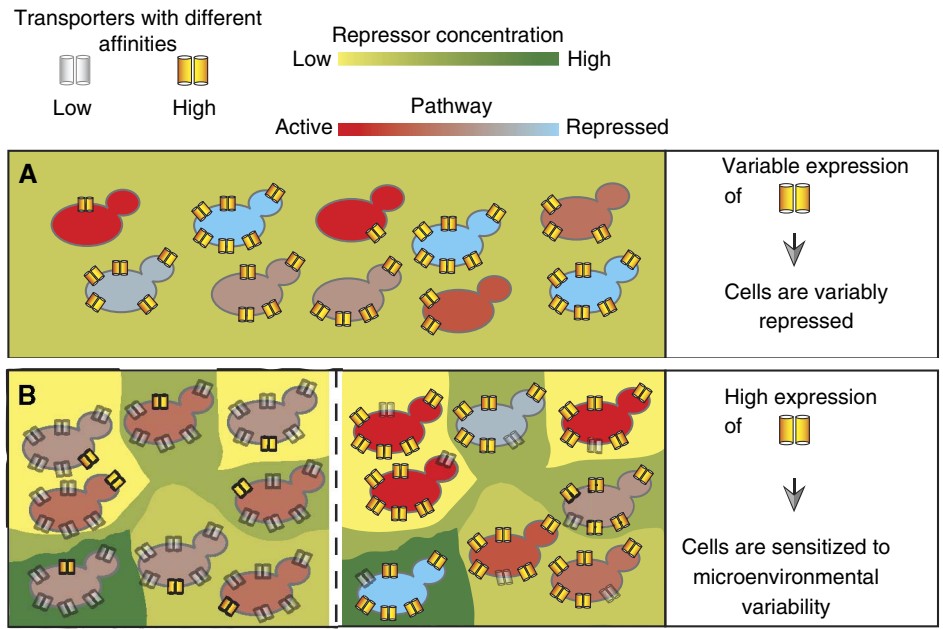

**Figure 7** Two distinct scenarios involving transmembrane transporters in the modulation of cellular individuality. (**A**) Cell–cell variability in expression of a transmembrane transporter generates a diversity of pathway activation. The pathway considered here is repressed by the molecule internalized via the transporter (such as methionine, internalized via Mup1p and repressing the pathway of sulfur amino-acid metabolism). Cells are colored according to the level of pathway activation, which is low in cells expressing the transporter at high levels because the repressor is better internalized. (**B**) In this case, transport of the repressive molecule is achieved via various transporters with different affinities. The expression level of one particular transmembrane transporter does not necessarily correlate with pathway activation in the cell. However, if a high-affinity transporter is overrepresented at the cell surface, cells may respond rapidly to microenvironmental variation in extracellular repressor concentration. This can in turn generate cell–cell variability in pathway activation. Left, a genotype with low expression of such a high-affinity transporter, such as BY expressing MUP1 at low levels. Right, a genotype with high expression of this transporter, such as RM.

relevant to further discoveries or applications. This is particularly important for traits related to industrial production yields: the effect of an introgressed locus can be tested across small-scale production processes to determine its potential usefulness before investing the effort to identify the causative gene. Introgression mapping is therefore very attractive in the context of organisms offering fast meiotic cycles.

It is remarkable that the two genes identified as ePTLs both coded for a transmembrane transporter. Deletion experiments on *PHO84*, another transmembrane transporter, also affected cell–cell variability. Transmembrane transporter genes were already pointed at by a previous study because they displayed higher cell–cell variability in expression than other classes of genes. In particular, *MUP1* itself was ranked among the 'noisiest' plasma membrane transporter genes, with 4.7 times higher cell–cell expression variability than average (Zhang *et al*, 2009). If the *MUP1* protein product was the only (or the major) yeast transporter for methionine, then the repressive molecule would be better internalized in cells with high *MUP1* expression. Variability in *MUP1* expression would therefore generate variability in the methionine-responsive pathway (Figure 7A). However, methionine can be imported into yeast cells by at least seven distinct transporters (Menant *et al*, 2006). Unless cell–cell variability in the expression of these transporters is coordinated, which is not expected given their different modes of regulation (Menant *et al*, 2006), expression variabilities of the seven transporters probably compensate one another. Our results show that *MUP1* causes cell–cell variability of the methionine pathway by increasing its

expression level in all cells of the population (Figure 7B and Supplementary Figure S2). How does this happen? One possibility is that high *MUP1* expression confers susceptibility to local variations in methionine concentration. The Mup1p transporter has higher affinity for methionine than all other transporters (Isnard *et al*, 1996). When abundant at the cell membrane, it probably allows cells to rapidly respond to dynamical changes of extracellular concentrations of methionine. In contrast, if low-affinity transporters predominate, cells may not be as responsive. Although we cultured cells in rather large liquid volumes (typically 4 ml) with shaking, the concentration of methionine in the immediate surroundings of each cell was neither constant nor strictly homogeneous. It is therefore possible that the RM promoter variant of *MUP1* generates variability by increasing the sensitivity to environmental fluctuations.

Our initial study established cell–cell variability in gene expression as a complex genetic trait but only indirectly, because only one ePTL had been identified and it corresponded to the artificial *ura3Δ0* mutation (Ansel *et al*, 2008). The work presented here fully demonstrates the polygenic architecture of cell–cell variability in gene expression, with the mapping of four additional ePTLs and the identification of two causative natural sequence variants. The fact that genotypes conferring high/low cell–cell variability segregate in natural populations is very important because it makes variability itself an evolvable trait. Alleles at a PTL are potentially subjected to selection, and this may happen in various ways. For example, selection may result from another, deterministic effect of the locus. Alternatively, at least theoretically, it may

also result from the benefit of elevated cell–cell variability in certain environmental conditions, especially in unpredictable environments. As mentioned in the Introduction, various studies have shown that phenotypic diversification among clonal cultures can be beneficial upon stressful conditions, which is considered to be a possible bet hedging strategy of adaptation. Experiments and simulations both showed that high cell–cell variability is unfit in fixed environments, and also in dynamic environments that change rapidly because the spontaneous switching rate between phenotypes may then be too slow (Acar *et al*, 2008; Stomp *et al*, 2008). In this latter case, a reactive strategy based on efficient sensors of environmental changes is predicted to win (Kussell and Leibler, 2005). However, whether natural genotypes conferring cellular individuality have been selected for in nature because of their benefit in slowly fluctuating environments remains to be demonstrated.

Here we show that several laboratory yeast strains acquired an *erc1* frameshift mutation that reduces cell–cell variability of *MET17* expression. Laboratory strains are maintained without selection pressure and across extreme population bottlenecks (propagation of single-cell colonies). This breeding habit results in 'wild-type' reference laboratory strains having accumulated slightly deleterious mutations that would likely be counter-selected in nature (Yvert *et al*, 2003; Warringer *et al*, 2011). The *erc1* frameshift mutation in BY likely results from such a genetic drift, as it suppresses two predicted C-terminal helices that are conserved across wild species. Accordingly, the observed low Ka/Ks ratio suggests negative selection on the protein sequence in the wild. The *ERC1* example therefore suggests that a PTL can originate from a loss of selective pressure on a protein sequence. Interestingly, the effect of this mutation is to reduce cell–cell variability in the methionine synthesis pathway as compared with the wild allele. This does not imply that *ERC1* is under selection for maintaining elevated variability. What this suggests is the following: let $\Delta F_{var}$ be the fitness cost of maintaining cell–cell variability in *MET17* expression, and let $\Delta F_{erc1}$ be the fitness cost of altering the C-terminal part of the Erc1p protein, then $\Delta F_{var} < \Delta F_{erc1}$. In other words, if elevated variability in *MET17* expression impaired fitness dramatically (very high $\Delta F_{var}$), then cells would be more fit with a modified Erc1p sequence and altered *ERC1* alleles should be found in the wild, which is not the case. Thus, some level of cell–cell variability in MET17 expression is maintained by *ERC1* in the wild, either because it is beneficial (which remains to be determined) or because suppressing it would be too costly.

It is more difficult to make evolutionary considerations on *MUP1*. The sequence polymorphism of interest is noncoding, and various combinations of the BY/RM SNPs can be found in wild alleles. Identifying which SNP or combination of SNPs affect MET17 expression variability would require a systematic and dedicated set of additional experiments. It is nonetheless worth noting that its protein sequence is highly conserved and under apparent negative selection (Ka/Ks = 0.035 between *S. cerevisiae* and *S. paradoxus*). Thus, the previously suggested positive selection for expression variability acting on this gene (Zhang *et al*, 2009) evolved within the constraint of maintaining the protein sequence unchanged.

The fact that the *pho84Δ* mutation had different effects when introduced into the BY or RM haplotype of the locus is interesting. It suggests that other polymorphisms of the locus are in genetic interaction with *PHO84* to modulate $P_{met17}$-GFP single-cell expression distribution. It can also be partly explained if a differential activity of BY and RM alleles of *PHO84* affects $P_{met17}$-GFP variability. In this case, deletion of the gene causes the removal of different molecular activities when done in the BY or in the RM context of the locus. We ruled out the implication of the L259P polymorphism, but different activities could result from another nonsynonymous SNP (D55N) or from noncoding ones. Identifying the precise genetic determinants of this effect would enable studying the mechanism involved and, in particular, whether it relates to phosphate transport, PKA activation or both.

Three of the four ePTLs mapped modified the expression variability of $P_{met17}$-GFP but not of $P_{act1}$-GFP. This specificity is consistent with the previous observation of coordinated levels of cell–cell variability among functionally related genes (Stewart-Ornstein *et al*, 2012). Although additional PTL examples are needed before drawing a general conclusion, our results suggest that natural PTLs affect cell–cell variability on specific pathways rather than pleiotropically. This modularity is likely advantageous for adaptation: some genotypes may increase cell–cell variability in a specific stress response or nutritional sensing without perturbing other physiological regulations. This way, various traits may be diversified independently in a population of cells, and if variability in one trait is beneficial for adaptation, it can be fixed while maintaining homogeneity of other traits.

Perhaps the most interesting evolutionary aspect of finding PTL sequence variants in environmental sensor genes is that it connects two previously opposed strategies of adaptation: anticipation by phenotypic diversification (probabilistic bet hedging) versus high reactivity to environmental changes (deterministic response) (Kussell and Leibler, 2005). The *MUP1* and *ERC1* sequence variants identified here act on both aspects. They improve the responsivity of cells to environmental changes in a deterministic way, and they increase cell–cell variability of *MET17* expression in a probabilistic way. Thus, mutations in environmental sensor genes may provide natural populations with a way to tune both adaptive strategies simultaneously.

# Materials and methods

## Strains and primers

Yeast strains and DNA oligonucleotides used in this study are listed in Supplementary Tables S2 and S3, respectively.

## Flow cytometry

Yeast cultures (synthetic medium, 2% glucose, 30 °C) and flow cytometry acquisitions were performed as previously described (Ansel *et al*, 2008). An overnight culture was diluted to $OD_{600} = 0.1$ in medium supplemented with 1 mM methionine (repressive condition) and grown for 3 h. Cells were then resuspended in medium containing 50 μM methionine, or divided in aliquots that were resuspended in a range of methionine concentrations (0, 5, 20, 50, 100, 150 and 200 μM). Cells were then grown for 2 h and analyzed on a FACSCalibur cytometer (BD Biosciences). Analysis of flow cytometry data was done using R (www.r-project.org) and the *flowCore* package of Bioconductor (www.bioconductor.org). GFP fluorescence (FL1

values) was corrected for cell size and granulometry by conditioning FL1 on Log(FSC) and Log(SSC), as previously described (Ansel et al, 2008). Cell–cell variability was calculated here as the CV (s.d. divided by mean) of the transformed intensities.

## Introgression mapping

Introgression mapping was initiated in our previous study, where partial genotyping of two strains (GY159 and GY174) that retained high CV values but carried only ~6.25% of the RM11-1a genome revealed the implication of ura3 as a QTL of cell–cell variability in the expression of $P_{met17}$-GFP (Ansel et al, 2008). Here we pursued the introgression by backcrossing these strains further with BY-derived strains. At every meiosis step, a spore displaying high CV and similar mean, as compared with BY, was chosen among 30–40 spores. Three backcrosses of GY174 led to GY768 (X7), and three backcrosses of GY159 led to GY769 (Z7). A third introgression lineage was initiated from GY601, an RM11-1a-derived strain harboring wild-type URA3. GY601 was backcrossed 7 times with BY derivatives, leading to strain GY793 (W7). This way, all three strains GY768, GY769 and GY793 had a wild-type URA3 gene, had ~1–2% of their genome originating from RM11-1a independently and displayed increased CV but unaffected mean expression as compared with BY. Genomic DNA of these three strains and of RM11-1a was extracted as previously described (Ansel et al, 2008) and subjected to whole-genome resequencing using paired-end sequencing on Illumina Genome Analyzer IIx. The sequencing reads from 4 samples (RM11-1a, W7, X7 and Z7) were aligned to S. cerevisiae S288c genome (http://www.yeastgenome.org) using novoalign V2.06.09 (http://www.novocraft.com) with parameters -rRandom -Q 75. SNPs were inferred by comparing the frequencies of each nucleotide at certain positions to the frequencies obtained from a previous resequencing of a BY strain. We obtained a list of 42 794 RM11 SNPs, which identified 11 regions that were ePTL candidates for the control of cell–cell variability of $P_{met17}$-GFP expression (Table I). To validate or reject these regions as ePTLs, we crossed one more time the introgressed strains with a BY derivative (either GY51 or GY172), isolated random spores, analyzed them by flow cytometry and genotyped them at the locus of interest using restriction fragment length polymorphism (RFLP) PCR (Supplementary Table S3). For every cross, we used a linear regression to condition CV on mean based on the data of all spores, and we used the residuals as quantitative trait values of the spores. We then tested for linkage between this trait and genotype at the locus by splitting the spores in two groups based on their genotype, and applying the Wilcoxon Mann–Whitney test on the null hypothesis of no trait difference between the two groups. Finally, to quantify the effect of each validated PTL, we constructed strains GY915, GY919, GY926 and GY927 that had retained the PTL locus but no other loci from RM11-1a, as determined by RFLP markers (Supplementary Table S3). We quantified by flow cytometry the expression of $P_{met17}$-GFP in replicated cultures of these strains across a range of methionine concentrations. During all experiments, the BY-derived strain GY172 or GY51 was also cultured and acquired in parallel for quantitative comparison.

## Assay on $P_{act1}$-GFP cell-cell variability

To test the pleiotropy of ePTLs, a synthetic $P_{act1}$-GFP construct flanked with BamHI sites was purchased from GeneCust Europe (Dudelange, Luxembourg) and cloned into the BamHI site of the pHO-poly-KanMX4-HO plasmid (Voth et al, 2001). A NotI fragment of this plasmid was transformed into strain BY4713 (Brachmann et al, 1998) to generate strain GY975 where the $P_{act1}$-GFP reporter was integrated at the HO locus. This strain was then crossed with strain GY930 (derived from GY919), random spores were isolated, analyzed by flow cytometry in the exact same conditions as above and genotyped at ePTL8 using RFLP PCR (Supplementary Table S3). Similarly, strain GY935 was crossed with GY975 and spores were analyzed by flow cytometry and genotyped at ePTL7 using RFLP PCR (Supplementary Table S3); strain GY512 was crossed with GY975, spores were analyzed by flow cytometry and genotyped at DST1 by PCR; and strain BY4719 was crossed with GY975, spores were analyzed by flow cytometry and

their URA3 locus was typed on −URA plates. The pleiotropy of ePTL13 was tested using strain GY1156, which harbors the same synthetic $P_{act1}$-GFP construct integrated at the HO locus except for the KanMX4 marker, which was replaced by TRP1. This strain was crossed with GY915 and random spores were analyzed by flow cytometry and genotyped at the ePTL using RFLP markers (Supplementary Table S3).

## Fine mapping of ePTL8

The diploid strain GY939 from which recombinant spores of the chrVIII interval were obtained was constructed as follows. Strain GY919 was crossed with BY4716 to obtain strain GY930, bearing the introgressed locus but lacking the $P_{met17}$-GFP-NatMX4 construct. To inactivate the COX6 gene by homologous recombination, the cox6Δ::KanMX4 locus from a strain bought from EUROSCARF was amplified with primers 1H29 and 1H30 and transformed into GY930, leading to strain GY931. Independently, a EUROSCARF strain thr1Δ::KanMX4 derived from the BY(S288c) background was crossed with GY172 to obtain GY929, harboring both the thr1 mutation and the $P_{met17}$-GFP-NatMX4 reporter cassette. GY929 was mated with GY931 to obtain GY939. This diploid was sporulated on potassium acetate plates, and random spores were plated on synthetic medium lacking threonine and supplemented with 3% glycerol as the sole carbon source. Spores were genotyped at markers within the introgressed locus in order to map recombination events.

## Sequence analysis

Protein sequences of ERC1 were downloaded from SGD (http://www.yeastgenome.org/), aligned using ClustalW with default parameters, and the alignment was manually corrected. The Ka/Ks ratio was computed by first aligning the coding sequence of RM11-1a (S. cerevisiae) with sequence MIT_Spar_C37_10276 (S. paradoxus) using ClustalW, and then applying the average method over multiple models implemented in the KaKs_Calculator version 1.2 software (Zhang et al, 2006). The same was done to compare the RM11-1a sequence with sequence MIT_Sbay_c47_10440 (S. bayanus). The promoter sequences of MUP1 were downloaded from SGD, aligned with ClustalW and the alignment was manually corrected.

## Allele replacements

To perform allele replacements, we first constructed a centromeric plasmid, pALREP, harboring the KlURA3 selectable marker, and where the CEN-ARS replicative sequence was flanked by LoxP sites. This way, the plasmid can be used in a replicative version for high-efficiency cloning using homologous recombination in yeast, and then, after Cre-mediated excision of CEN-ARS, as an integrative version for targeted mutagenesis. To construct this plasmid, we first removed one LoxP site from plasmid pUG-KlURA3 (Delneri et al, 2000) by digestion with SacII and religation. Resequencing of the resulting plasmid indicated that the SacII site was destroyed by the process. We then amplified the LoxP-CEN-ARS-LoxP segment from plasmid pDS163 (Sinclair and Guarente, 1997) by PCR with primers containing StuI and NheI sites and the amplicon was cloned into the PvuII and XbaI sites of the modified pUG-KlURA3 plasmid.

The ERC1 locus was amplified from RM11-1a genomic DNA by Herculase polymerase (Agilent Technologies Inc.) using primers 1I12 and 1I13. This amplicon was cotransformed in a yeast ura3Δ0 strain with pALREP linearized at HpaI for cloning by homologous recombination. The resulting plasmid was verified by restriction mapping and transferred in bacteria carrying the 705-Cre expression system (Gene Bridges GmbH), where expression of the Cre recombinase was transiently induced. Plasmid molecules were extracted and retransformed in E. coli to select a plasmid (pGY251) where the CEN-ARS sequence had been excised. Strain GY246 (Ansel et al, 2008) was transformed with pGY251 linearized at NcoI and two URA$^+$ transformants (GY1008 and GY1009) were chosen on the basis of proper integration as determined by AvrII RFLP PCR amplified with 1H89 and 1H90 primers. GY1008 and GY1009 were transferred to 5FoA plates to select for loss of KlURA3, leading to strains GY1023 and

GY1024, respectively. Final genotypes were verified by AvrII RFLP PCR amplified with 1H89 and 1H90.

Similarly, the *ERC1* locus was amplified from BY genomic DNA by Herculase polymerase (Agilent Technologies Inc.) using primers 1I12 and 1I13, cloned in pALREP, and the CEN-ARS sequence of the resulting plasmid was excised as above to generate plasmid pGY250. This plasmid was linearized at *Nco*I and integrated into strain GY53 by selection on − URA plates. Transformants were verified by AvrII RFLP using primers 1H89 and 1H90 and one strain harboring the desired BY allele (GY1010) was replicated on 5FoA plates for loss of KlURA3. Two strains resulting from independent recombination events, GY1019 and GY1020, were selected based on proper genotype at the AvrII RFLP.

A similar approach was applied to replace the *MUP1* promoter. A MUP1 fragment was amplified from BY genomic DNA using Herculase (Agilent Technologies Inc.) with primers 1I32 and 1I29 and cloned by homologous recombination in pALREP linearized at *Hpa*I. The CEN/ARS sequence of the resulting plasmid was excised by transient expression of the Cre recombinase in *E. coli* as above to produce plasmid pGY256. Strain GY53 (Ansel *et al*, 2008) was transformed with pGY256 linearized at *Eco*RI. URA+ transformants were genotyped by PCR and sequencing to verify proper integration. One of them was then plated on 5FoA plates to select for loss of *KlURA3*. Four strains resulting from independent pop-out events (GY1205, 1206, 1207 and 1208) displayed the BY genotype at all SNP positions of the MUP1 promoter, as determined by high-resolution melting PCR with primers 1I80 and 1I81, and sequencing of PCR product obtained with primers 1G84 and 1G85.

The *ATR1* promoter was replaced similarly by cloning into pALREP, a PCR fragment amplified from RM genomic DNA with primers 1J10 and 1J11, and removing the CEN/ARS sequence by Cre/lox excision. The resulting plasmid pGY303 was linearized at *Bsa*BI and integrated into strain GY246. URA+ transformants were genotyped by PCR (primers 1J12, 1J13) and one of them (GY1329) was replica-plated on 5FoA plates to select for loss of *KlURA3*, which led to strain GY1330.

### MUP1 promoter activity

To obtain a convenient plasmid for GFP reporter activity, we modified pGY8 (Ansel *et al*, 2008) by adding in it the lox-CEN/ARS-lox sequence of pALREP and changing the *MET17* promoter by the *PGK* promoter directly upstream the GFP coding sequence. This plasmid, pGY252, was then modified by homologous recombination to replace the *PGK* promoter by the *MUP1* promoter amplified from RM genomic DNA using Herculase DNA polymerase with primers 1I35 and 1I36. The resulting plasmid (pGY257) was then treated by the 705-Cre system as above to excise the CEN/ARS region, generating an integrative plasmid (pGY260) that was linearized by *Nhe*I within the HIS3 cassette and integrated in strain BY4716. This produced three independent transformants, GY1002, GY1003 and GY1004. Similarly, the *MUP1* promoter of BY was amplified and cloned into pGY252, the resulting plasmid was modified to become integrative (plasmid pGY261) and transformed into BY4716 to produce three independent transformants, GY1005, GY1006 and GY1007.

### ATR1 promoter activity

Similarly, the *ATR1* promoter was amplified from BY and RM genomic DNA using primers 1I98 and 1I99 and cloned into pGY252 by homologous recombination to produce plasmids pGY296 and pGY297, respectively. Their CEN/ARS sequence was removed by Cre/Lox excision, and the resulting plasmids were linearized at *Nhe*I and integrated into strain BY4716 to produce strains GY1325 and GY1326 (independent transformants) as well as GY1327 and GY1328 (independent transformants).

### Knockout analysis

The *atr1Δ::KanMX4* allele was amplified from EUROSCARF strain Y16516 using primers 1J06 and 1J07 and transformed into strain GY1284 to produce strain GY1302, which was then crossed with GY172 to produce GY1309. Similarly, the *pho84Δ::KanMX4* allele was amplified from EUROSCARF strain Y16524 using primers 1J08 and 1J09 and transformed into strain GY1284 to produce GY1303, which was then crossed with GY172 to produce GY1310.

## Supplementary information

## Acknowledgements

We are grateful to Martine Collart, Magali Richard and Valérie Robert for critical reading of the manuscript, Soizic Riche for technical help, Leonid Kruglyak for the PHO84-L259P strain, the Pôle Scientifique de Modélisation Numérique (Lyon, France) for computer resource, Sandrine Mouradian and SFR Biosciences Gerland-Lyon Sud (UMS3444/US8) for access to flow cytometers and thermal cyclers, Franck Vittoz for the design and construction of prototype labware equipments, developers of R, Bioconductor, Lyx and Ubuntu for their software and three anonymous reviewers for their comments. This work was supported by Grant ANR-07-BLAN-0070 from the Agence Nationale de la Recherche France (to GY), by the European Research Council under the European Union's Seventh Framework Programme FP7/2007–2013 Grant Agreement no. 281359 (to GY) and by the National Institutes of Health (to LMS).

*Author contributions:* SF and AL: performed experiments and analyzed the data; HB-D, EM and AB: performed experiments; WW: analyzed genome re-sequencing data; LMS: contributed reagents; GY: conceived and designed the study, analyzed the data, wrote analysis codes, interpreted results and wrote the paper.

## Conflict of interest

The authors declare that they have no conflict of interest.

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
