## [Review Process File · Molecular Systems Biology]

Natural sequence variants of yeast environmental sensors confer cell-to-cell expression variability

Steffen Fehrmann, H el ene Bottin-Duplus, Andri Leonidou, Esther Mollereau, Audrey Bartheleix, Wu Wei, Lars M. Steinmetz, Ga el Yvert

Corresponding author: Ga el Yvert, Ecole Normale Sup erieure de Lyon, CNRS, Universit e Lyon 1

Review timeline:

Submission date:	14 February 2013
Editorial Decision:	25 March 2013
Revision received:	17 July 2013
Editorial Decision:	14 August 2013
Revision received:	27 August 2013
Accepted:	06 September 2013

Editor: Maria Polychronidou

Transaction Report:

1st Editorial Decision

25 March 2013

Thank you again for submitting your work to Molecular Systems Biology. We have now heard back from the three referees who agreed to evaluate your manuscript. As you will see from the reports below, the referees find the topic of your study of potential interest. They raise, however, substantial concerns on your work, which should be convincingly addressed in a revision of the manuscript.

One major point that was raised by reviewer #2 and should be carefully addressed, refers to the need to map all the alleles conferring variability in order to support the statement that all determinants of variability have been identified.

Moreover, the reviewers point out that the manuscript should be carefully re-written in order to become accessible to a broad audience, while certain issues regarding the terminology and citation of previous studies need to be addressed.

If you feel you can satisfactorily deal with these points and those listed by the referees, you may wish to submit a revised version of your manuscript. Please attach a covering letter giving details of the way in which you have handled each of the points raised by the referees. A revised manuscript will be once again subject to review and you probably understand that we can give you no guarantee at this stage that the eventual outcome will be favorable.

Referee reports:

Reviewer #1:

This is a thorough analysis of causal genetic variation underlying phenotypic variation of expression of a specific promoter in clonal populations. The results and analysis support the interpretation and the conclusions are sound. This is an important advance in the field.

A couple of minor issues:

The description of the flanking markers in the cross for narrowing down one of the ePGL regions is not clear in the text though the figure is clear.

It isn't fair to state that they have made a complete determination of the ePGLs for phenotypic variation as there are clearly other loci in BY modifying the effects in this cross.

These ePGLs are just like QTLs in that they can have non-additive interactions and depend on the genetic background as well as the environment, therefore the introductory material defining strict lack of increased CV due to epistasis and G x E interactions may be confusing.

Reviewer #2 :

The study by Fehrmann et al. is interesting. The authors describe the isolation of naturally occurring alleles that confer variability for expression of MET17. The fact that the alleles are specific rather than pleiotropic is of interest and the study may have implications for our understanding of the evolution of anticipatory strategies for adaptation.

The study is however not completely convincing in all aspects and is also only half-finished in others. Furthermore, the way that the paper has been written up can be vastly improved.

1. Fine-mapping has only occurred for 2 of the 4 alleles. Although the difficulties of fine-mapping are appreciated, this is not very satisfying. It also does not fit with the claims that the authors have identified all the causes of variability. The authors should fine-map the other alleles. There are several strategies for this.

2. Once fine-mapped, the authors should make two additional strains. The RM strain with the variability alleles replaced by the alleles from BY. A BY strain with the corresponding alleles from RM. These should then be tested appropriately for CV of expression and would be a much more convincing test of whether all sources of variability have been identified. The current experiments have been done with very large regions.

3. Even with this additional work, the study will remain anecdotal to an extent. This can be improved by experiments aimed at testing hypotheses about the mechanism of individual alleles for conferring increased variability. For example, the second identified allele is a cis-acting mutation, presumably affecting expression through the action of a transcription factor or nucleosome. If the authors reasoning is correct this means that other cis-acting mutations should be readily found that confer the same properties. This would go a little way at least to confirm mechanism. Perhaps the authors can come up with other experiments. Proper testing of the underlying mechanisms would certainly raise the level of the paper.

4. The paper is currently very expansively written and presented and additional experiments should easily fit into the paper if the authors were to present more concisely. The authors currently use the first four figures to come to four unprecisely mapped alleles for a single trait. This could be put into a single figure, making room for the additional analyses suggested above.

5. Great improvements in readability can be made. All sections are very lengthily written up and in many cases quite miss the point of trying to inform a general readership. This remark applies to lengthy individual sentences which require frequent rereading by the (genetically) uninitiated to be understood. I had to look up the meaning of many terms to understand what the authors were trying

to convey. It would be best if the authors rewrite the manuscript bearing a general readership in mind and then have a general molecular biologist underline anything that is not immediately clear. The current paper is not easily accessible to a wide body of scientists, although the subject is of general interest.

Reviewer #3 :

The authors conduct a very nice study on genetic control of genes controlling single cell variability in methionine related gene expression within yeast. This provides the identification of several new alleles controlling variability in this phenotype. This is an excellent illustration of how genes known to affect the mean phenotypic value can have alleles that are specific for the variance component of the same trait. This is something that is rarely acknowledged and the authors do an excellent job of showing this. The materials and methods are excellent and I have no issues with those. There are some areas of significant concern in the citations and quantitative genetics terminology that I feel should be fixed to properly position this excellent paper in the broader literature.

There are some issues with the introduction. In one instance, the authors cite a manuscript of theirs in preparation which is not appropriate for a final published paper. Additionally, the authors do not cite the example in Arabidopsis where the authors conducted genome wide stochastic transcriptomic variation and cloned an underlying locus (1). This citation would appear to be key in the introduction when they describe loci shown to control variability especially as the Yang and Kemkemer only associate genes with variability and do not validate them as was done in (1) providing the second known ePGL. Additionally, this citation also showed partitioning of variability between phenotypes and cis/trans as argued in this paper.

The authors suggest that a PGL changes the probability that an individual expresses a trait, however in quantitative genetics, trait is the item such as height while phenotype is the level of the trait, i.e. 2 m tall. Do the authors really mean that a PGL controls if the organism has height at all or a specific height. If the later, then it should be the value of the trait or phenotype. If they do mean trait, then they should specify that they are talking about traits with a threshold response.

Additionally if a PGL is merely a genetic polymorphism altering stochastic variance how can it not be subject to epistasis or GxE? It is merely a gene and all genes are likely to show GxE or epistasis. This was found in numerous maize studies on yield stability as cited by the author as well as in the Drosophila citations. Especially as the authors show that stacking ePGLs that should add to 0.071 only causes a 0.049 which is the definition of epistasis.

The sentences "The second limitation is the detection power of PGL. Much larger samples are needed to obtain a statistical description of the trait than to estimate its mean value." Is not fully accurate. The authors mean that to obtain a similar power on describing variability requires more samples.

In the results section at the top of page 9, why don't the authors just state that random drift and a bottleneck could lead to different partitioning of loci across the lineages? That would be more efficient. The authors already knew from other papers that variability was a polygenic trait so this section is largely a straw man argument. It is a great control on finding the real loci but doesn't need the A versus B aspect.

(1) JimÈnez-GÙmez, J. M., Corwin, J. A., Joseph, B., Maloof, J. N., and Kliebenstein, D. J. (2011). Genomic analysis of QTLs and genes altering natural variation in stochastic noise. *PLoS Genet* 7, e1002295.

RESPONSE TO ALL REVIEWERS:

We thank the reviewers for their helpful comments.

To address some of the points raised by reviewer #2, we performed experiments on candidate genes of the locus Chromosome 15 position 563573 to 703057, which we called ePGL15 in the initial version of the manuscript. When doing so, we realized that strain GY920, which carries the RM-introgressed version of this locus, did not reproducibly show elevated variability of Pmet17-GFP expression. We inspected laboratory notebooks and realized that two consecutive and independent errors had led us to wrongly call this locus an ePGL. First, the p-value of the validation/rejection linkage test was not calculated correctly. It included a few data points corresponding to replicates of the parental strains, which are not independent observations. Removing these replicates and keeping only the independent values from segregants gave $p = 0.4$. We verified all other p-values of Table 1, and they are correct. The second error occurred much later, by another experimenter, when analyzing strain GY920. The analysis had been done simultaneously as strain GY919 and there is a risk that all samples were in fact GY919, which reproducibly shows elevated variability. Locus 'ePGL15' was therefore a false positive and the revised text has been corrected accordingly. We apologize for these errors, and we sincerely thank reviewer #2 for requesting additional experiments that allowed us to correct them in the revisions.

In addition to the revisions described below, we have changed the term 'Probabilistic Genetic Locus (PGL)' by 'Probabilistic Trait Locus (PTL)' to better relate to the non-deterministic nature of the phenotype and not of the genotype.

POINT-BY-POINT RESPONSE:

Reviewer #1 :

This is a thorough analysis of causal genetic variation underlying phenotypic variation of expression of a specific promoter in clonal populations. The results and analysis support the interpretation and the conclusions are sound. This is an important advance in the field.

Thank you.

A couple of minor issues:

The description of the flanking markers in the cross for narrowing down one of the ePGL regions is not clear in the text though the figure is clear.

We rephrased this description as follows: "As shown on Figure 4A, the region is flanked by two genes that can be used as selective markers. *THR1* codes for a homoserine kinase essential for threonine biosynthesis (Mannhaupt et al, 1990). The *COX6* gene codes for a subunit of cytochrome C oxidase essential for respiration (Gregor & Tsugita, 1982). Thus, both *THR1* and *COX6* functional genes are required for growth on a synthetic medium lacking threonine and with glycerol as the sole carbon source (Gly, Thr-)."

It isn't fair to state that they have made a complete determination of the ePGLs for phenotypic variation as there are clearly other loci in BY modifying the effects in this cross.

We agree and apologize for this overstatement which has been corrected in abstract, end of introduction, results (page 10), and discussion (page 16).

These ePGLs are just like QTLs in that they can have non-additive interactions and depend on the genetic background as well as the environment, therefore the introductory material defining strict lack of increased CV due to epistasis and G x E interactions may be confusing.

We thank reviewers 1 and 3 for pointing to the confusion. The purpose of this part was to conceptually distinguish between hidden deterministic modifiers from the genetic background or the environment, and a truly probabilistic effect of the locus in a controlled genomic and environmental context. This is now better explained in the revised text: "Note that many QTL have deterministic effects that manifest only in specific genetic or environmental contexts, and therefore only in some individuals. In this case, the locus does not need to be called a PTL because its effect is not inherently probabilistic. Finally, the probabilistic effect of a PTL may be modulated by non-additive interactions with the genetic background or the environment, just as the deterministic effect of a QTL."

Reviewer #2 :

The study by Fehrmann et al. is interesting. The authors describe the isolation of naturally occurring alleles that confer variability for expression of MET17. The fact that the alleles are specific rather than pleiotropic is of interest and the study may have implications for our understanding of the evolution of anticipatory strategies for adaptation.

The study is however not completely convincing in all aspects and is also only half-finished in others. Furthermore, the way that the paper has been written up can be vastly improved.

1. Fine-mapping has only occurred for 2 of the 4 alleles. Although the difficulties of fine-mapping are appreciated, this is not very satisfying. It also does not fit with the claims that the authors have identified all the causes of variability. The authors should fine-map the other alleles. There are several strategies for this.

The claim that all causes of variability were identified has been removed from the text. Since fine-mapping the ERC1 locus took us over a year, and since other ePTL were not flanked by convenient markers to do a similar strategy, we did not plan additional fine-mapping within the allowed time for revisions. However, we performed various experiments on candidate genes. First, we studied knock-outs of PDR5 and SER1 genes which allowed us to reject the locus on chromosome XV (please see above, response to all reviewers). Second, we studied deletion strains of ATR1 and PHO84, either in the standard BY context or in the context where the PTL locus is of RM genotype. Third, we compared the activity of BY and RM version of the ATR1 promoter region. Fourth, we replaced the allele of ATR1 in the BY strain by the RM sequence. And fifth, we tested a functional non-synonymous SNP of PHO84. These efforts were not sufficient to fine-map ePTL13 but they clearly ruled out the implication of ATR1 and they revealed a functional impact of PHO84 on the modulation of Pmet17-GFP variability, which further reinforces

the role of transmembrane transporter genes in this variation. These experiments are presented in Figure 6 and described on pages 15-16 of the revised manuscript.

2. Once fine-mapped, the authors should make two additional strains. The RM strain with the variability alleles replaced by the alleles from BY. A BY strain with the corresponding alleles from RM. These should then be tested appropriately for CV of expression and would be a much more convincing test of whether all sources of variability have been identified. The current experiments have been done with very large regions.

We tried several times to replace the endogenous MUP1 promoter of BY by the RM allele but the few transformants obtained did not display the expected genotype. We were therefore not able to make an exhaustive comparison of allelic combinations. This is a further reason why we removed the claim that all sources have been identified.

3. Even with this additional work, the study will remain anecdotal to an extent. This can be improved by experiments aimed at testing hypotheses about the mechanism of individual alleles for conferring increased variability. For example, the second identified allele is a cis-acting mutation, presumably affecting expression through the action of a transcription factor or nucleosome. If the authors reasoning is correct this means that other cis-acting mutations should be readily found that confer the same properties. This would go a little way at least to confirm mechanism. Perhaps the authors can come up with other experiments. Proper testing of the underlying mechanisms would certainly raise the level of the paper.

Regarding MUP1 expression difference, we agree that a detailed mechanistic investigation would explain how this gene is upregulated in RM. This would need to study promoter polymorphisms one by one and in combination, test their effect in mutant backgrounds and validate the implication of transcription factors or nucleosomes by ChIP. This wealth of experiments probably go beyond the scope of the study, which message is to demonstrate the genetic properties of variability as a complex trait. We agree that *cis*-acting mutations represent interesting candidates to have the same *trans*-ePTL effect as MUP1. In response to this comment, we now present a detailed analysis of the *cis*-acting polymorphic region of the ATR1 gene where *cis*-eQTL effect is demonstrated and *trans*-ePTL effect is clearly rejected. This shows that *cis*-acting variation does not necessarily confer the same properties as the one seen for MUP1.

4. The paper is currently very expansively written and presented and additional experiments should easily fit into the paper if the authors were to present more concisely. The authors currently use the first four figures to come to four unprecisely mapped alleles for a single trait. This could be put into a single figure, making room for the additional analyses suggested above.

The first four figures of the initial submission are now grouped into two figures only. This made room for the new figure 6 describing the results obtained on ATR1 and PHO84.

5. Great improvements in readability can be made. All sections are very lengthily written up and in many cases quite miss the point of trying to inform a general readership. This remark applies to lengthy individual sentences which require frequent rereading by the (genetically) uninitiated to be understood. I had to look up the meaning of many terms to understand what the authors were trying to convey. It would be best if the authors rewrite the manuscript bearing a general

readership in mind and then have a general molecular biologist underline anything that is not immediately clear. The current paper is not easily accessible to a wide body of scientists, although the subject is of general interest.

We have shortened many sentences, especially in the results section, and we hope that the revised text is more accessible now.

Reviewer #3 :

Natural sequence variants of yeast environmental sensors confer cell-to-cell expression variability. Yvert, et al.

The authors conduct a very nice study on genetic control of genes controlling single cell variability in methionine related gene expression within yeast. This provides the identification of several new alleles controlling variability in this phenotype. This is an excellent illustration of how genes known to affect the mean phenotypic value can have alleles that are specific for the variance component of the same trait. This is something that is rarely acknowledged and the authors do an excellent job of showing this. The materials and methods are excellent and I have no issues with those.

Thank you.

There are some areas of significant concern in the citations and quantitative genetics terminology that I feel should be fixed to properly position this excellent paper in the broader literature.

There are some issues with the introduction. In one instance, the authors cite a manuscript of theirs in preparation which is not appropriate for a final published paper.

This paper is now published at BMC Systems Biology, and the reference has been included. The study can be accessed at: <http://www.biomedcentral.com/1752-0509/7/54>.

Additionally, the authors do not cite the example in Arabidopsis where the authors conducted genome wide stochastic transcriptomic variation and cloned an underlying locus (1). This citation would appear to be key in the introduction when they describe loci shown to control variability especially as the Yang and Kemkemer only associate genes with variability and do not validate them as was done in (1) providing the second known ePGL. Additionally, this citation also showed partitioning of variability between phenotypes and cis/trans as argued in this paper.

We totally agree that we should have cited this reference in the introduction. This is now corrected in the revised manuscript.

The authors suggest that a PGL changes the probability that an individual expresses a trait, however in quantitative genetics, trait is the item such as height while phenotype is the level of the trait, i.e. 2 m tall. Do the authors really mean that a PGL controls if the organism has height at all or a specific height. If the later, then it should be the value of the trait or phenotype. If they do mean trait, then they should specify that they are talking about traits with a threshold response.

We changed "expresses a trait at a given value" by "expresses a given trait value" in the definition of PTL. We also changed "trait" by "phenotype" or by "trait value" when referring to the quantitative value of the trait.

Additionally if a PGL is merely a genetic polymorphism altering stochastic variance how can it not

be subject to epistasis or GxE? It is merely a gene and all genes are likely to show GxE or epistasis. This was found in numerous maize studies on yield stability as cited by the author as well as in the Drosophila citations. Especially as the authors show that stacking ePGLs that should add to 0.071 only causes a 0.049 which is the definition of epistasis.

We thank reviewers 1 and 3 for pointing to the confusion. The purpose of this part was to conceptually distinguish between hidden deterministic modifiers from the genetic background or the environment, and a truly probabilistic effect of the locus in a controlled genomic and environmental context. This is now better explained in the revised text: "Note that many QTL have deterministic effects that manifest only in specific genetic or environmental contexts, and therefore only in some individuals. In this case, the locus does not need to be called a PTL because its effect is not inherently probabilistic. Finally, the probabilistic effect of a PTL may be modulated by non-additive interactions with the genetic background or the environment, just as the deterministic effect of a QTL."

The sentences "The second limitation is the detection power of PGL. Much larger samples are needed to obtain a statistical description of the trait than to estimate its mean value." Is not fully accurate. The authors mean that to obtain a similar power on describing variability requires more samples.

We have changed this sentence by "At similar power, detecting differences in variance or other high-order moments of a statistical distribution requires larger samples than detecting differences in mean."

In the results section at the top of page 9, why don't the authors just state that random drift and a bottleneck could lead to different partitioning of loci across the lineages? That would be more efficient. The authors already knew from other papers that variability was a polygenic trait so this section is largely a straw man argument. It is a great control on finding the real loci but doesn't need the A versus B aspect.

This has been corrected. The revised text now reads " However, given our previous observation of polygenicity, this could simply be due to random drift and bottlenecks. At every generation, we chose one spore out of only about 40. If, for example, six loci account for the elevated variability of RM, then only one spore out of $2^6=64$ would harbor all genetic determinants. In this case, at any of the seven selection steps, measuring expression CV in only 40 spores may have led us to focus on a spore that retained only a subset of the loci of interest. And this subset may differ from one lineage to another. We therefore considered all remaining nine regions as ePTL candidates..."

2nd Editorial Decision

14 August 2013

Thank you again for submitting your work to Molecular Systems Biology. We have now heard back from the two referees who agreed to evaluate your manuscript. As you will see from the reports below, the reviewers acknowledge that you have satisfactorily addressed most of their major concerns. However, reviewer #2 points out that in its current form, the manuscript is still not easily accessible to a broad audience. Therefore, we would like to ask you to simplify the text as much as possible throughout the manuscript. Moreover, we would strongly recommend the inclusion of a "Didactic Box", (namely an illustration and a short explanatory text), in order to explain in intuitive language the concept of probabilistic trait locus (PTL) in comparison to the conventional QTL.

We have included a few suggestions for changes regarding the abstract, which you can find in the attached document. Additionally, Figure 7 would benefit from a few modifications in order to make the key points immediately visible to the readers (i.e. a color coded bar depicting the increasing pathway activation, inclusion of text helping the readers go through the two different scenarios only by looking at the scheme, modification of the scheme in order to make clear to the reader that in the upper panel there is only one transporter involved while in the lower one several transporters). Finally we think that the discussion should be streamlined, as it is relatively long in its current form. Thank you for submitting this paper to Molecular Systems Biology.

Referee reports:

Reviewer #1 :

I am happy with the responses to my comments as well as to the other reviewers.

Reviewer #2 :

The authors have not addressed all the issues raised. It is also a pity that they have paid little attention to the remarks about writing in a way that would make their study be appreciated by a wider audience. One of the sentences in the abstract is a good example: "These results describe the complex genetic architecture of cell-to-cell variability in gene expression and illustrate that a deterministic cis-eQTL can have a probabilistic effect in trans." I think that only a small percentage of scientists who should read the paper, will actually do so and of these only a fraction will understand sentences like the one highlighted. I think that it is a pity (because it is fine work), but this is up to the editor or the authors to decide.

2nd Revision - authors' response

27 August 2013

Thank you for processing our manuscript entitled "Natural sequence variants of yeast environmental sensors confer cell-to-cell expression variability" through a fast second round of peer-reviewing. Please find enclosed a revised version of the paper. Following the comment of reviewer #2 and your recommendations, we have applied the following changes to improve clarity and accessibility to a broader audience:

- We have added a didactic box (referred to as 'Box I' in text and legend), with a figure and text illustrating the distinction between a PTL and a QTL.

- We modified Figure 7 to highlight the key points. Text was included to explain scenarios A and B, color bars were added, the presence of different transporters in scenario B is now shown, and the micro-environmental variation was added as a color-coded background.

- We have simplified and clarified the abstract and text at many places (indicated in green).

- We streamlined the discussion by simplifying the text and removing several sentences that were redundant.

We are sincerely excited by the perspective to publish our work in *Molecular Systems Biology* and we are looking forward to hearing from the journal.